

# In-situ cloud ground based measurements in Finnish sub-Arctic: Intercomparison of three cloud spectrometers.

Konstantinos-Matthaios Doulgeris[1], Mika Komppula[2], Sami Romakkaniemi[2], Antti-Pekka Hyvärinen[1], Veli-Matti Kerminen[3] and David Brus[1]

[1] Finnish Meteorological Institute, PO Box 503, FI-00101, Helsinki, Finland
[2] Finnish Meteorological Institute, PO Box 1627, FI-70211, Kuopio, Finland
[3] Institute for Atmospheric and Earth System Research/Physics, Faculty of Science, University of Helsinki, Finland

*Correspondence to*: Konstantinos M. Doulgeris (konstantinos.doulgeris@fmi.fi)

**Abstract.** Continuous, semi-long term, ground based, in-situ cloud measurements were conducted during the Pallas Cloud Experiment (PaCE) in 2013. The measurements were carried out in Finnish sub-Arctic region at Sammaltunturi station (67°58′N, 24°07′E, and 560 m a.s.l.), the part of Pallas Atmosphere - Ecosystem Supersite and Global Atmosphere Watch

(GAW) programme. The main motivation of the campaign was to conduct in-situ cloud measurements with three different cloud spectrometer probes and perform an evaluation of their ground based setups. Therefore, we mutually compared the performance of the Cloud and Aerosol Spectrometer (CAS), the Cloud Droplet Probe (CDP) and the Forward Scattering Spectrometer Probe (FSSP-100), (DMT, Boulder, CO, USA). We investigated how different meteorological parameters affect each instrument operation and quantified possible biases and discrepancies of different microphysical cloud properties. Based

on obtained results we suggested limitations for further use of the instruments in campaigns where focus is on investigating aerosol cloud interactions. Measurements in this study were made by Finnish Meteorological Institute owned instruments and results concern their operation in sub-Arctic conditions with frequently occurring super-cooled clouds.

Measured parameter from each instrument was the size distribution and additionally we derived the number concentration, the effective diameter, the median volume diameter and the liquid water content. A complete intercomparison

between the CAS probe and the FSSP-100 and additionally between the FSSP-100 and the CDP probe was made and presented. Unfortunately, there was not sufficient amount of common data to compare all three probes together due to operational problems of the CDP ground setup in sub-zero conditions. The CAS probe that was fixed to one direction lost significant number of cloud droplets when the wind direction was out of wind iso axial conditions in comparison with the FSSP-100 and the CDP which were both placed on a rotating platform. We revealed that CAS and FSSP-100 had good agreement in deriving

sizing parameters (effective diameter and median volume diameter from 5 to 35 µm) even though CAS was losing a significant amount of cloud droplets. The most sensitive derived parameter was liquid water content which was strongly connected to the wind direction and temperature.



# 1 Introduction

Clouds and their interaction with aerosol particles provide some of the greatest uncertainties in predictions of climate change (Boucher et al., 2013). Therefore, in situ measurements of clouds play a key factor for further investigation of the aerosol cloud interaction area. Many of the climatically important cloud properties (e.g. albedo, precipitation rate and lifetime) depend, among other factors, on the number concentration of aerosol particles and on their chemical composition (Komppula et al., 2005; Lihavainen et al., 2008). Measuring the cloud size distribution is an important parameter for identification and description of clouds; their microphysical properties (Pruppacher and Klett, 1977; Rosenfeld and Ulbrich, 2003), and their lifetime (Albrecht 1989; Small et al., 2009).

One major category of instruments that it is commonly used for in situ cloud measurements are known as cloud spectrometers (e.g. Knollenberg, 1976; Dye and Baumgardner, 1984; Wendish, 1996; Baumgardner et al., 2001; Lance et al., 2010; Baumgardner et al., 2014). Cloud spectrometers are single particle counters that use the forward scattering, usually with the angles between 4 and 12º of a laser beam to detect and classify in different size bins individual particles. The main theory used for the particle sizing based on the scattering of light from single particles is the Lorenz - Mie theory (Mie, 1908). Several experiments were conducted with those instruments; they mainly cover: laboratory (e.g. Wagner et al., 2006; Smith et al., 2015; Nichmann et al., 2017), ground based (e.g. Mertes et al., 2001; Bukarel et al., 2002; Henning et al., 2002; Eugster et al., 2006; Lihavainen et al., 2008; Loyd et al., 2015) and airborne measurements (e.g. Knollenberg et al.,1981; Heymsfeld et al., 2004; Bromwich et al., 2012; Johnson et al., 2012, Jones et al., 2012, Briswick et al., 2014; Korolev et al., 2014; Petäjä et al., 2016; Webke et al., 2016; Vogt et al., 2017; Faber et al., 2018)

In addition to above mentioned experiments, many studies were done to quantify biases, uncertainties and limitations of cloud spectrometers while they were used in measurement campaigns. Uncertainties were usually a result of different meteorological conditions. Baumgardner (1983) presented a comparison of five water droplet instruments, included the axially scattering spectrometer (ASSP) and the forward scattering spectrometer probe (FSSP). He concluded that scattering probes had an accuracy of 17% in number concentration and size measurements. A full description and evaluation of the FSSP was made by Baumgardner et al., (1985) investigating coincidence and dead-time losses and by Baumgardner et al., (1990) investigating time response and laser inhomogeneity limitations. Baumgardner et al., (1989) made a calibration of the FSSP during the airborne Antarctic zone experiment and set further limitations to be applied during the data analysis of this project. A similar study from Baumgardner et al., (1992) was conducted for the FSSP during the airborne arctic stratospheric expedition where an improved forward scattering spectrometer probe, the FSSP-300, was developed and introduced. Wendisch (1998) presented a quantitative comparison of ground based FSSP with a particle volume monitor. He stated that FSSP can be considered as an excellent microphysical sensor in continental, stratiform or cumuliform clouds with mostly small drops, however he noticed some discrepancies in the liquid water content, especially when cloud droplets larger than 25 µm were considered. Gerber et al., (1999) performed and evaluated ground based measurements of liquid water content using also a FSSP and a particle volume monitor. They observed large discrepancies too and stated that the FSSP overestimate liquid water



content for large cloud droplets due to the inertial concentration effect. Burnet and Brenquier (1999) validated the droplet spectra and the liquid water content using five instruments including the FSSP, the fast FSSP and the CDP. Burnet and Brenquier (2002) investigated further in detail only the FSSP to address the instrumental uncertainties and limitations of them. Coelho et al., (2005) made a detailed discussion for FSSP-100 in low and high droplet concentration measurements with a

proposed correction for coincidence effects. Lance et al., (2010) calibrated the CDP and presented a full description of the instrument performance in laboratory and in-flight conditions. Baumgardner et al., (2011) summarized airborne systems for in situ measurements of aerosol particles, clouds and radiation that were currently in use on research aircraft around the world including cloud spectrometer probes. Febvre et al., (2012) highlighted the possible effects of ice crystals in FSSP measurements.  Spiegel et al., (2012) made a thoroughly analysis of wind velocity and wind angle impacts at the Junfraujoch

comparing the Fog droplet spectrometer to others instruments. One more evaluation regarding cloud ground based measurements which taking into consideration the wind direction was made by Guyot et al., (2015) at the Puy-de-Dôme observatory between seven optical sensors including a FSSP, a fast FSSP, a fog monitor and two CDP probes. Authors showed a good agreement in sizing abilities of the instruments but observed possible discrepancies in number concentration values, fact that also affected the liquid water content values. Several developments of the in-situ measurement systems were reviewed

and summarized with respect to their strengths, weaknesses, limitations and uncertainties by Baumgardner et al., (2017). The progress in performing in-situ cloud measurements was clearly observed along with developments.

In this work, we focused on the intercomparison of three cloud spectrometer probes as they were used during the PaCE 2013. The FMI research station (Sammaltunturi) located in northern Finland is considered as an ideal place to perform in-situ low level cloud measurements, especially during autumn, when the station is usually inside a cloud about 50% of the time.

There, along with the FSSP – 100 and the CDP which are classic instruments for in-situ cloud measurements, we were also using the Cloud, Aerosol and Precipitation Spectrometer (CAPS) probe (part of this instrument is the CAS probe) with a purchased inhalation system. CAPS' worldwide unique setup allows us semi-long term (in orders of months) observations when compared to short-term (orders of hours) airborne measurement. The exact set of measurements limitations for each cloud probe that are presented in this work will help us to conduct a detailed further analysis of microphysical cloud properties

and their interactions with aerosol during all PaCE campaigns. The previous PaCE campaigns, already resulted in series of publications on experimental observations and modelling studies (e.g. Komppula et al., 2005 and 2006; Lihavainen et al., 2008 and 2010; Kivekäs et al., 2009; Anttila et al., 2009 and 2012).

A description of the measurement site and the instrumentation as it was installed could be found in section 2.1 and 2.2. Subsequently, in section 2.3, the procedure we followed during data analysis is presented. In section 3, the inter-comparison

of the instruments and how they were influenced by the meteorological parameters are presented. Finally, in section 4, we summarized our main results and conclusions in order to set limitations and made recommendations for the data analysis of the three instrument ground based setups during future studies in sub-Arctic environment.



## 2. Methods

### 2.1 Measurement site description

Measurements were conducted during autumn 2013, in the Finnish sub-Arctic region at Sammaltunturi station (67°58′N, 24°07′E, 560 m a.s.l.) which is a part of the Pallas Atmosphere – Ecosystem Supersite hosted by the Finnish Meteorological Institute. The station is also part of Global Atmosphere Watch (GAW) programme. Sammaltunturi station is located on the top of a treeless hill. A full detailed description of the site can be found in Hatakka et al., (2003). Autumn was chosen as the best period to run the campaign due to the high chances that the station will be inside a cloud. This allows us to conduct in-situ cloud measurements. All the meteorological parameters were continuously measured by the Vaisala FD12P weather sensor. During our previous Pallas Cloud Experiments (PaCE) clouds microphysical properties and aerosols physico – chemical properties and their interactions were measured and investigated (e.g. Lihavainen et al., 2008; Hyvärinen et al., 2011; Anttila et al., 2012; Collaud Cohen et al., 2013; Jaatinen et al., 2014; Lihavainen et al., 2015; Raatikainen et al., 2015; Gérard et al., 2019). During PaCE2013, our main motivation was to focus on inter-comparison of in-situ cloud properties measured with three different cloud probes, their evaluation and mutual benchmarking regarding PaCE campaigns.

### 2.2 Cloud instrumentation

During PaCE 2013, to perform in-situ measurements of cloud droplets, we used three instruments originally developed for airborne measurements, but tailored in co-operation with the manufacturer (DMT, USA) for ground-based measurements. The Cloud, Aerosol and Precipitation Spectrometer probe (CAPS), the Cloud Droplet Probe (CDP) and the Forward Scattering Spectrometer Probe (FSSP-100, hereafter called as FSSP for simplicity). All three of them were installed on the roof top of the measurement site as it is described below in details and share similar measurement technique. The basic concept is that they use the forward scattering of a laser beam for the detection and sizing of individual particles. Then, using Mie theory (Mie, 1908), they calculate the size of the particle from the intensity of the scattered light.

Only data of the Cloud and Aerosol Spectrometer (CAS) probe were used regarding to the CAPS. CAS is one part of the Cloud, Aerosol and Precipitation Spectrometer probe (CAPS, DMT) (0.51-930 µm) which is an instrument that is widely used on airborne measurements for investigating the microphysical properties of clouds (e.g. Baumgardner 2001; Baumgardner et al., 2011). The CAPS probe includes two more instruments, however they are not comparable with the FSSP and the CDP probe: the Cloud Imaging Probe (CIP) and the Hotwire Liquid Water Content Sensor (Hotwire LWC). CAS measures smaller particles (0.51 µm to 50 µm) and relies on light-scattering. Particles scatter light from an incident laser at a wavelength of 680 nm, a sample area of 0.24 mm² and collecting optics guide the light scattered in the 4° to 12° range into a forward-sizing photodetector. The intensity of light is measured and used to infer the particle size. Backscatter optics also measure light intensity in the 168° to 176° range, which allows determination of the real component of a particle's refractive index for spherical particles. The droplets are then classified into 30 size bins. CAS was operational from October 15th until November 28th. It was installed and fixed on Sammaltunturi station roof. It was heading to the main wind direction of the station (southwest, ~ 225°). For the instruments' installation we used the manufacturer pylon (height 0.3 m) (same as it is used for



CAPS airborne measurements). The whole system was fixed on horizontal metallic circle (D = 0.28 m). The metallic circle was attached to a vertical metallic bar (height 0.3 m) part of a square metallic stand (0.7 m x 0.7 m) (see Fig1a). As a result CAPS had a total height of 0.6 m from the point of the roof it was installed and 4.5 m from the ground. In our setup, CAPS had its own tailored inhalation system, a high flow pump (Baldor – Reliance, USA), which worked continuously. The pump

was connected with the CAS probe with 1.14 m long of 0.07 m inner diameter hose. The hose was connected to a triple branch (three parts with I.D. = 0.12) through a 0.12 m to 0.05 m reducer. The triple branch connected the CAS probe through the hose with the high flow pump. The other parts of the branch connected the pump with the CIP through 2 different hoses (L=1.52 m I.D. = 0.12 m). In addition, a stepped CAS inlet (funnel shape reducer I.D. = 0.12 to I.D. = 0.05 m) was attached over the CAS inlet tube (for detailed description please Fig. S1 and Fig. S2 in the Supplement).The probe air speed (PAS) inside CAS was

checked daily with a digital thermo anemometer (model 471, Dwyer Inc.) to secure that the flows inside the instrument remained similar. This was done through a small hole near the end of CAS probe outlet and beginning of 0.05 m hose and in a position such that the anemometer inserted into hole was just in the middle of the CAS probe outlet (hose diameter). In cases when PAS changed, data were corrected accordingly. During this campaign PAS values ranged from 17 to 23 ms$^{-1}$. The calibration of the instrument was done at DMT and also at the Finnish Meteorological Institute before and after the campaign

using glass beads and polystyrene latex spheres (PSL) standards.

The Forward Scattering Spectrometer Probe (FSSP, 1.2-47 μm, model SPP-100, DMT), initially manufactured by Particle Measuring Systems (PMS Inc., Boulder CO, USA) is a widely used cloud probe for measuring droplet size distribution (Brenquier 1989). It shares the same measurement principle with the CAS probe and measures the light scattered in the 4° to 12° range with a laser of wavelength 633 nm and a sample area of 0.414 mm$^2$. Droplets were classified into 40 size bins.

During PaCE, the FSSP was installed and placed on a rotating platform, so that the inlet was always heading against the wind direction. The rotating platform was a horizontal metallic base (0.7 x 0.1 x 0.4 m) with a metallic fin fixed at the back of it. This setup ensured that the instrument would follow the wind direction continuously. The rotating platform was placed on a similar squared metallic stand we used also in CAPS setup, but with a higher metallic vertical bar (L = 0.6 m, I.D. = 4 cm). Thus, the instrument had a total height of 0.6 m from the point of the roof it was installed and 5.5 m from the ground. During

FSSP installation on the rotating platform, we wanted to prevent the full rotation of the probe which could be dangerous for the cable wiring and safety of the instrument. For this reason, a vertically metallic bar (0.3 m, D. = 0.6 cm) along with two horizontal bars (L = 0.25 m, D = 0.6 cm) were installed (northeast ~ 60°) and they served as a brake (Fig. 1 b). A custom inhalation system with high flow ventilator was employed through the instruments' inlet to secure that the air speed would remain constant (for detailed description please Fig. S3 in the Supplement). In addition, the PAS inside the FSSP tube was

checked daily with the digital thermo-anemometer (model 471, Dwyer Inc.). The PAS during the campaign was ~ 10 ms$^{-1}$ which lead to an air speed of ~ 36 ms$^{-1}$ inside the inlet due to necking inside the inlet's mouth from I.D. = 3.8 cm to I.D. = 2.0 cm. This value was used for further data processing. FSSP was installed and operational from September 14$^{th}$ until November 28$^{th}$ 2013. The instrument was calibrated at DMT, USA before and on site after the end of the campaign.



The third instrument that we used was the cloud droplet probe (CDP, 3-50 μm, Droplet Measurement Technologies) (Lance et al., 2010). Similar to the previous instruments it uses the same principle and measures the light scattered in the 4° to 12° range. The laser beam had wavelength of 658 nm and sample area of 0.3 mm². The CDP classified droplets into 30 size bins. It was placed next to the FSSP also on a rotating platform to continuously head against the wind direction (Fig. 1 b). The metallic platform covering the instrumental electronics consists of fixed part (0.4 x 0.4 x 0.3 m) at the bottom and on top of that the rotating part (0.4 x 0.4 x 0.1 m) having the probe itself on top of it. The rotating part is equipped with a large fin to keep the inlet towards the wind (for detailed description please Fig. S4 in the Supplement). The instrument had a custom inlet with an external pump to secure a constant PAS (14 ms⁻¹). The CDP was installed and was operational from September 25th 2013 until November 28th 2013. It was calibrated at Finnish Meteorological Institute, Kuopio unit, before the campaign and after the campaign on site using glass beads and PSL standards.

All three instruments were using anti ice heaters as they were deployed by the manufacturer. The external parts of the setup (rotating platforms and inhalation systems) were not using additional heating system. The instruments were installed in a horizontal position and placed close to each other on Sammaltunturi roof. The CDP and FSSP were installed next to each other (approx. 0.5 m) and they had a horizontal distance of ∼ 10 m and vertical distance of ∼1 meter to the CAS probe. All the probes' parameters are presented in Table 1. During the campaign a routine was consistently followed. The cloud probes functionality was daily checked visually. Ice and snow accumulation could fully or partially block the probes' inlets and affect the flows. In addition, ice and snow could also accumulated in parts of the roof and affect the probes measurements. For this reason, all three cloud probes needed periodical cleaning. When the station was inside a cloud and sub-zero temperatures were observed, the cleaning procedure of the probes during the day was repeated twice or more times per day (if needed).

## 2.3 Data handling and processing

The presence of a cloud was estimated by the cloud droplet counts measured with all the cloud probes, the visibility and relative humidity measurements at the site. As a cloud event we defined the situation when there were droplet counts (considering cleaned dataset) measured by the cloud probes more than 30 continuous minutes, the horizontal visibility was less than 1000 meters and the relative humidity was ∼ 100%.

From each cloud probe we obtained the cloud droplet size distribution. For the intercomparison of the probes we had to eliminate some size bins of the CAS and the FSSP probe in order to use similar size range in each case. The CAS probe, using the PADS software (Droplet Measurement Technologies Manual, 2011), derives the number concentration ($N_c$), the liquid water content ($LWC$), the median volume diameter ($MVD$) and the effective diameter ($ED$). The same parameters were derived using the following equations, since we have used old software PACS 2.2 (Droplet Measurement Technologies) for data acquisition of the FSSP-100 and the CDP probe:

Number concentration, ($N_c$, cm⁻³) was calculated from the division of the total number of sized particles $N$ with the sample volume $V_s$ (cm⁻³)

$$N_c = \frac{N}{V_s},$$
(1)





where $V_S$ was defined as

$$V_S = \text{PAS} \times t \times A \tag{2}$$

where PAS is the probe air speed (ms$^{-1}$), $t$ is the time of the sampling interval and $A$ is sample area (mm$^2$) defined as the height of the laser beam (mm) multiplied by the length of the laser beam within the depth of field (DOF, mm). On instrument that records probe time as CAS and FSSP, the sampling interval is calculated by subtracting the previous instance's probe time from that of the current instance. On the CDP the sampling interval is assumed to be the designated sample rate. All three probes were setup to sample at 1s (1Hz).

Liquid water content, ($LWC$, gm$^{-3}$) is the mass of liquid water per unit volume of air and it was calculated using the following equation

$$LWC = \sum_i^n LWC_i \tag{3}$$

where

$$LWC_i = c_i 10^{-12} \frac{\pi}{6} m_i^3 \tag{4}$$

$m_i$ is the midpoint of its bin and calculated as

$$m_i = \frac{b_i + b_{i+1}}{2} \tag{5}$$

and $c_i$ are the droplets concentrations per bin (m$^{-3}$).

The factor $\frac{\pi}{6} m_i^3$ in equation indicated that we assume that the particle is a sphere. Another assumption that was made is that water has a density of 1 g cm$^{-3}$.

Median volume diameter ($MVD$, µm) indicates the droplet diameter which divides the total water volume in the droplet spectrum such that half the water volume is in smaller drops and half is in larger drops and is derived by a linear interpolation with respect to the (i+1) bin as

$$MVD = b_{i^*} + \left(\frac{.5 - cum_{i^*-1}}{pro_{i^*}}\right)(b_{i+1} - b_{i^*}) , \tag{6}$$

where $pro_i = \frac{LWC}{LWC_i}$ is the proportion of the spectrum $LWC$ that falls in the $i$-th bin and

$cum_i = pro_1 + \cdots + pro_i$ is the cumulative proportion of the spectrum $LWC$ that falls in the first $i$ bins and

$i^*$ is the smallest value of $i$ such that $cum_i > 0.5$.

This interpolation gives an accurate estimation in comparison with the half point of $b_{i^*}$ and $b_{i+1}$. The second component of the equation scales the amount summed to $b_{i^*}$ according how close $b_{i^*}$ and $b_{i+1}$ each were to 0.5.

Effective diameter ($ED$, µm), which is the ratio of $LWC$ to the optical cross sectional area of droplets of a sample droplet spectrum by the following equation:

$$ED = \frac{\sum_{i=1}^n p_i r_i^3}{\sum_{i=1}^n p_i r_i^2} 2 \tag{7}$$



where $n$ is the number of sizing bins, $p_i$ the particle counts for bin $i$ and $r_i$ the mean radius in µm of bin $i$.

All our instruments were calibrated in laboratory and /or on site. The calibration of all three instruments was done for size measurements but not for $N_c$ measurements. Also, we should take into account the fact that the instruments faced extreme conditions during the whole campaign, in terms of frequent changes to wind direction and speed and sub-zero temperatures. Those meteorological conditions could possibly lead to unexpected performance.

The sizing accuracy for cloud spectrometers has been estimated as 20% and the concentration accuracy as 16% (Baumgardner, 1983; Dye and Baumgardner, 1984; Baumgardner et al., 2017). The major factors that are usually considered for possible biases in data analysis are coincidence, dead time losses and changing velocity acceptance ratio (*VAR*). Coincidence events occur when more than one droplet is registered by an instrument at the same time resulting in multiple droplets artificially measured as one droplet. Changing velocity acceptance ratio (*VAR*) is a result from the fact that only a part of a laser beam is used to calculate the sampling volume because drops passing the laser beam near the edges are undersized. Lance et al. (2012) showed that at ambient droplet concentrations of 500 cm⁻³, at least 27% undercounting and 20%–30% oversizing bias were observed for CAS. However, during PaCE 2013 droplet number concentrations we observed to reach maximum of 200 cm⁻³ and in majority of cases less than 100 cm⁻³. Due to those low number concentration values we do not take coincidence, dead time losses and *VAR* uncertainties into consideration in this analysis. Liquid water content (*LWC*) as it was derived from the CAS probe has a significant uncertainty of 40% according to Droplet Measurement Technologies Manual. FSSP uncertainties, limitations and corrections have been several times reported in literature (Baumgardner 1984; Baumgardner et al., 1985; Baumgardner and Sportwart, 1990). Febvre et al. (2012) find out that the uncertainty of the FSSP in derived effective diameter (*ED*) and derived liquid water content (*LWC*) are 3µm and 30% regarding mixed phase clouds. Lance et al. (2012) reported for the CDP probe importance of the coincidence errors even if the number concentrations are as low as 200 cm⁻³ resulting in 25% undercounting error and 30% oversizing error due to coincidence.

During the campaign, measurements were performed with 1 Hz acquisition frequency for all three probes. During the data analysis, minute averages from each cloud probe were calculated when the station was inside a cloud. As we previously highlighted, there were cases when the flows of the cloud probes were blocked. This situation was also visible in raw data. Such cases were cleaned out from all datasets for the final analysis. A typical example case of probe freezing was observed on November 3rd, 2013. The CDP was completely clogged with ice, see Fig. 2, where its size distribution is presented. However, in order to find probe freezing cases a closer look in $N_c$ was carefully done for the whole data set. As an example, Fig. 3 depicts the $N_c$ of each cloud probe during the same day along with the meteorological parameters. There, the sudden decrease in droplet number concentration of the CAS probe from 12:00 and a sudden increase in $N_c$ just before was a clear sign of probes inlet freezing. This behavior in $N_c$ was observed due to the opening of the probe inlet was getting smaller (from the accumulation of ice) which resulted in a raising PAS. During data evaluation we considered the PAS was constant. This led to an underestimation of the PAS which explain this abnormality in the $N_c$. The same behavior could be also seen for the CDP after 16:00.



## 3   Results
### 3.1   Overview of results for PACE 2013

During PaCE 2013, FSSP, CAS, and CDP were installed for the total of 1824, 1080, and 1560 hours respectively, see
Table 1 for installation dates. During the campaign, the station was inside a cloud about 664 hours (36.5%). During this period,
the temperature ranged from -12.0 to 10.2 °C with averaged temperature -1.9 (std 5.1) °C and the wind speed average was 6.8
(std 2.9) ms$^{-1}$. The dominant wind direction was ~ 220 ° but there were winds and clouds from all directions. Regarding data
coverage, the FSSP and the CAS probe showed good performance with ~ 500 hours (75%) and ~ 220 hours (67%) cleaned
data coverage respectively. The CDP probe performed significantly worse with ~ 108 hours (17%) cleaned data coverage. The
CDP worse performance was due to its frozen inlet or/and its rotation system during night and encountered frequent operation
problems especially at sub-zero temperatures.

### 3.2   Example cases with all three probes in operation.

Firstly, we present example cases when all three probes were operating. Those cases correspond to different wind
directions. This choice was made since we used two different approaches; two probes were installed on a rotating platform
(FSSP, CDP) and one probe was installed in a fixed direction (CAS). We provide the time series of the derived $N_c$, $LWC$, $ED$,
$MVD$ and the size distributions as they were measured by all three probes along with selected meteorological parameters
(temperature, wind speed and wind direction).  For each case, the same size range for each probe was adopted. For this reason,
we eliminated 12 bins (ranging from 0.51 to 3 µm) of the CAS probe and 2 bins (from 1.2 to 3.5 µm) of the FSSP-100. The
final size ranges for probes comparison were then following: the CAS 3 to 50 µm, the FSSP 3.5 to 47 µm and the CDP 3 to 50
µm.

The above mentioned parameters are presented on November 12$^{th}$ from 15:00 to 18:00, see Fig. 4. The wind speed during
this period varied from 6 to 13.5 ms$^{-1}$ and temperature varied from -6.1 to -5.1 °C. Average wind direction was 226.5 (std 6.1)°,
which means that all three instruments were pointing to the same direction (Fig. 6e). As we noticed in Fig. 4a, this resulted to
a good agreement among all three probes in deriving $N_c$. The CDP operated without flaws in both its rotation and inhalation
system due to cleaning procedure of the instrument done just before this measuring period. In this case, the $LWC$ values were
in best possible agreement, see Fig. 4b. Derived sizing parameters $ED$ and $MVD$ are presented in Fig. 4c and d. and both of
them had good agreement.

The next example case took place during October, 29$^{th}$ from 12:15 to 15:00, see Fig. 5. The wind speed during this period
varied from 2.9 to 8.9 ms$^{-1}$ and temperature varied from -5.4 to 2.9 °C. Average wind direction was 141.2 (std 18.4)°, which
indicates that the wind direction was perpendicular to the CAS probe. (Fig. 5e). Here, the CAS probe significantly
undercounted $N_c$ (~ factor 5) compared to FSSP and CDP (Fig. 5a). The CDP was also operating with no malfunctions in its
rotation and inhalation system. $LWC$ as it was derived by the CAS probe was highly affected by a factor of about 7 due to its
losses in $N_c$ (Fig. 5b). Furthermore, as we can see in Fig. 5c and d, CAS ability to derive $ED$ and $MVD$ was not affected by $N_c$.





Especially, when we are comparing *ED* and *MVD* between CAS and FSSP, their difference is less than 20%. However, it was also interesting that even the CDP and the FSSP had a good agreement in droplet counts, they present some differences in the other derived parameters. Investigating in details their size distribution, we found that this was a result of different estimation in sizing. This can be clearly seen in Fig. 5f. In this case there was a shift in CDP sizing when compared with FSSP towards

smaller sizes about 5 μm in size range from 15 to 20 μm.

The last example day we present took place on October 25th, from 12:00 to 15:00, see Fig.6. The wind speed varied during this period from 4.4 to 9.9 ms⁻¹ and temperature from -5.8 to -5.3 °C. Average wind direction was 85.6 (std 4.8)°, which means that the probes were not facing the same direction (Fig. 6e). This explains why the $N_c$ of CAS was lower than the $N_c$ of FSSP by a factor of 2. The CDP measured the same $N_c$ of cloud droplets as CAS, something that was not really expected (Fig. 6a).

This is a typical example case we faced which indicates why the CDP counts were not trusted during sub-zero temperatures. Even if the CDP was not obviously clogged, when observed from the raw data, its rotation system was frozen so it could not follow the wind direction and we were not able to determine where exactly the CDP was pointing. Also, here, we can see that during the period the instruments were not facing the same wind direction, we observed large discrepancies in derived *LWC* (Fig. 6b). In Fig. 6c and 6d we can also see slight discrepancies in derived sizing parameters *ED* and *MVD*. We noticed a slight

shift in FSSP sizing towards bigger sizes compared to CAS, about 2.5 μm in size range from 7 to 10 μm and a slight shift in CDP sizing towards smaller sizes, about 2 μm in size range from 5 to 7 μm, as it is depicted in Fig. 6f clearly explains those differences.

**3.3  CAS and FSSP counting performance based on the wind direction.**

After investigating different example cases, we focused on how the change of the wind direction influenced the droplet counting ability of the instruments in different sectors of the wind rose. In this section, we concentrated only on the inter-comparison of the CAS probe (installed in a fixed direction) with the FSSP (was following the wind direction). The decision to first compare only CAS and FSSP setups was made because their parallel data coverage was the best (~ 243 hours of common cleaned data set). CDP was not used in this section due to only few common data with the CAS. The reason for that

was that from the date that the CAS was installed the CDP had operation problems. A detailed analysis regarding the CDP and its behaviour is presented later in section 3.5. To obtain as close as possible size range for both instruments we removed the first ten bins of the CAS and the first bin of the FSSP. As a result, we used the following size ranges; CAS - from 1.25 to 50 μm and FSSP - from 1.2 to 47 μm.

To estimate the possible losses of the cloud droplet counts from each cloud probe we used $N_c$ as a benchmark parameter.

Possible CAS sampling losses were investigated by calculating the aspiration efficiency as described in Spiegel et al., 2012. Our expected losses were ~5% for 20 μm, ~20% for 40 μm and 40% for 50 μm cloud droplets.  Averaged total $N_c$ values of cleaned data set as they were derived from the CAS probe and the FSSP were 39.8 (std 35.3) cm⁻³ and 44.1(std 26.9) cm⁻³ respectively. We divided the wind rose into 12 parts. This choice was made according to specific factors. First of all, we took





into consideration the crucial point orientation of CAS inlet when compares to actual wind direction since the CAS probe was fixed and installed (southwest of the station, ~ 225°). This point helped us to define the areas where the two instruments were performing in wind iso-axial condition. Secondly, we tried to secure that we will have enough amount of data in each part to increase the reliability of our results. Accordingly, the 360 wind rose was divided in to following sectors: 0 to 74, 75 to 94, 95

to 114, 115 to 154, 155 to 184, 185 to 199, 200 to 214, 215 to 235, 236 to 250, 251 to 265, 266 to 295 and 296 to 360°. Fig. 7 shows the ratio of $N_c$ of the CAS to the FSSP probes along with the percentage of observations in each of those sectors of the wind rose and the averaged $N_c$ values from both instruments. There, we can see that each of instruments had different counting performance in each sector. The best counting performance (ratio is close to 1) was found at two sectors (from 200 to 214° and 215 to 235°), where both probes were facing similar direction. On the other hand, when the wind direction was

perpendicular (115 to 154°) to CAS fixed direction, the ratio was found lower than 0.4. There the CAS probe undercounted a significant amount of cloud droplets (~ 60%). However, there were also cases where FSSP measured smaller $N_c$ compared to the CAS probe (sectors from 236 to 250° and from 251 to 265°). During those cases, FSSP was not actually freely moving because of the brake set up. Depend on the wind turns, FSSP could be left in wrong orientation for an unknown amount of time.  Inside those two sectors the CAS probe measured relatively high $N_c$ (~ 120cm$^{-3}$) in comparison to the other parts of the

wind rose (~ 50 cm$^{-3}$).

All wind sectors were further investigated to explain the biases in the performance of the two instruments. Firstly, a closer look (see Fig.8) is presented for two sectors (200 - 214, 215 - 235°) where the agreement was found the best according to $N_c$ ratio. For this reason, the wind rose sector from 200 to 235° was adopted as wind iso-axial conditions for the rest of this work. Results indicate that the agreement in both cases was good ($R^2$ = 0.78 and 0.62 with slope 0.65 and 0.50 respectively, Fig.8a

and b) and the maximum difference observed was ~ 30%. When $N_c$ as derived from CAS was more than 80 cm$^{-3}$, FSSP $N_c$ was about 25% lower. Temperature and wind speed in range of -11 to -1.4 °C and 1.6 to 13.8 ms$^{-1}$ were also tested for possible biases in wind iso-axial conditions and we found that they did not affected the ability of the probes to derive $N_c$. A more detailed look of how the two cloud probes measured in wind iso-axial conditions when the station was inside a cloud is presented in Fig. 8c and d, where the averaged number size distribution of the cloud droplets is shown. The CAS probe

measured more counts in sizes smaller than 7 μm (~ 3 counts / cm$^{-3}$ more than the FSSP at 1.4 μm and ~ 15 drops / cm$^{-3}$ more at 5 μm, in both cases difference in counts was less than 30%).  Also, we can see that the FSSP measured no droplets for sizes larger than 35 μm. Within the size range, meaning from 7 to 20 μm (area which usually represented the peak of the size distribution), the FSSP usually measured higher $N_c$. This difference could be up to 25% (~ 150 more counts / cm$^{-3}$). We have noticed also a slight shift in FSSP sizing towards bigger sizes, about 1.5 μm in size range from 7 to 10 μm. Those differences

in the counting efficiency of the two instruments explain the slight discrepancies we observed in $N_c$ even when they were measuring in wind iso-axial conditions.

In a similar way, all the remained sectors of the wind rose were investigated in detail to reveal more biases. In Fig. 9 we summarized the most representative cases. Fig. 9a shows the whole wind iso-axial conditions sector as it was defined





previously (200 – 235°) and ensures that there was good agreement ($R^2 = 0.70$ and slope 0.57). Fig. 9b shows that the CAS probe had more losses (factor from 3 to ~ 10) in $N_c$ when the wind direction was perpendicular to the CAS fixed direction, covering the sector from 115 to 154° ($R^2 = 0.32$ and slope 0.72). We also used observations when the wind direction ranged from 0 to 74° (Fig. 9c). There, due to the installation of the brake in FSSP' setup, an abnormality was created which clearly

affected FSSP' ability to operate properly. The agreement between the two instruments in this sector of the wind rose was found the worst of all cases ($R^2 = 0.08$ and slope 0.33). Finally (Fig. 9d), we used observations when the wind direction ranged from 95 to 114° in order to demonstrate one case when the wind direction was out of both, the wind iso-axial and perpendicular area. As expected, the CAS probe was affected by the wind direction. CAS was undercounting again when deriving $N_c$ (slightly less than in the case of perpendicular direction, $R^2 = 0.54$ and slope 0.64). Figure 10 presents the number size distributions for

the same cases to investigate further the counting ability of the two instruments and find out the size bins where the probes had the biggest difference in counting. For size range from 1.2 to 7 μm, both cloud probes behaved the same in all wind directions. In Fig. 10a (200 – 235°) we noticed that the number size distribution in wind iso-axial case had only some minor differences in sizing (slight shift in FSSP sizing towards bigger sizes, about 1.5 μm) that were expected as we mentioned in the previous paragraph. In Fig. 10b (115 to 154°), where the wind was perpendicular to the CAS probe we lost a significant number

(maximum losses in counts up to 75%) of droplets in the size range from 8 to 30 μm. In Fig. 10c (0 to 74°), where the FSSP faced operational malfunction due to its brake installation setup, it undercounted cloud droplets (maximum losses in counts up to 85%) for sizes larger than 11.8 μm. Finally, in Fig 10d (95 to 114°) we observed that the behaviour of CAS was affected by the wind direction in a similar way as it was found for the perpendicular case. However, in this case CAS lost fewer droplets (maximum losses in counts up to 45% for size range from 8 to 30 μm).

20        From the inter-comparison of the two instruments in each sector of the wind rose, a general benchmarking was created and it is presented in Fig. 11. According to our results we merged some of the wind sectors that we had initially created. As a result, we now have four sectors representing the wind rose; wind iso-axial conditions (from 200 to 235°), perpendicular conditions (from 115 to 154 and 296 to 360°), conditions between iso-axial and perpendicular (from 76 to 114, from 155 to 199 and from 236 to 295°) and the special case where the brake influenced the performance of FSSP (from 1 to 74°). To

summarize our results, we should highlight that the best agreement between the two cloud spectrometers was obtained in wind iso-axial conditions (from 200 to 235°, $R^2 = 0.60$) and it covered a cleaned data set of ~ 66 observation hours. The effect of wind direction on the CAS probes ability to measure $N_c$ was immediately noticed when the wind direction was out of the range of the wind iso-axial conditions. The agreement became slightly worst when the spectrometers were facing conditions that wind direction was between iso-axial and perpendicular ($0.46 \leq R^2 \leq 0.50$ for 76 to 114, 155 to 199 and 236 to 295° respectively,

~ 50% of total cleaned data set). The CAS probe performed the worst when the wind direction was perpendicular to the CAS installed direction ($R^2 = 0.32$ and 0.11 for 115 to 154 and 296 to 360° respectively) and represents ~ 40 observation hours.





### 3.4 Inter-comparison of CAS and FSSP - derived parameters *LWC*, *ED* and *MVD*.

In this section, we focused on investigating the derived parameters *LWC*, *ED* and *MVD*. First, a comparison of the liquid water content (*LWC)* for the two probes CAS and FSSP is presented. We only present measurements that were performed in wind iso-axial conditions, since the *LWC* was very sensitive to both changes in droplet $N_c$ and changes in shape of the number size distribution. The discrepancies we observed in droplet $N_c$ in sectors outside the wind iso-axial conditions caused a significant difference in total *LWC* at least by a factor of 5 or even more. We also noticed differences by factor of 15 especially when the wind direction was perpendicular to CAS fixed direction. Figure 12a shows that the agreement in *LWC* ($R^2$ = 0.34 and slope 0.53) between CAS and FSSP in iso-axial conditions was found worse than agreement of both probes in $N_c$. After investigating how different meteorological parameters contribute to apparent biases in more detail, we found that temperature was the main and only factor that affected the instruments ability to derive *LWC*. Accordingly, we divided our measurements in two temperature data sets. Measurements with temperature range from -11.1 to -4 and from -3.9 to -1.4 °C. Figure 12b presents the agreement for the first set of measurements, temperatures below -4 °C. Excluding the warmer temperature range, we obtained better agreement between the probes ($R^2$ = 0.78 and slope 0.82). On other hand, the second set of temperatures (from -3.9 to -1.4 °C) indicated that the two probes significantly disagreed ($R^2$ = 0.02 and slope 0.07). As we already explained in section 3.3, there was a slight shift in FSSP sizing towards bigger sizes, about 1.5 µm in size range from 7 to 10 µm. However, when applied the correction to FSSP sizing the resulting change in LWC was found marginal (about 0.7%).

Our main conclusion regarding the derived *LWC* was that the main factor affecting *LWC* values was the actual difference in the counts in each bin, especially when referring to larger droplets. Taking into account those limitations and biases in deriving *LWC* our final proposal is to use only *LWC* values from wind iso-axial conditions. In addition, the critical parameter should be the temperature of the cloud. This suggests that only derived *LWC* values for temperatures below -4 °C will be considered as acceptable and will be used for further analysis of this data set. However, even when we consider the best agreement the maximum difference in obtained *LWC* between CAS and FSSP could still be about 40%. In addition, we suggest the deployment of another *LWC* sensor, e.g. the particle volume monitor (PVM-100, Gerber 1999) during future campaigns in order to obtain another reference *LWC* values for inter-comparison in wide temperature range.

The final step to complete the inter-comparison between the CAS probe and the FSSP was to investigate their ability to derive two sizing parameters, the *ED* and the *MVD*. Both of them are significant to identify and evaluate the sizing performance of the cloud spectrometers (e.g. Stephens, 1978; Slingo and Schencker, 1982; Korolev, 1999; Mitchell et al., 2011). The cleaned dataset obtained from the whole wind spectrum plotted in different color scale for wind directions, temperature and wind speed is presented in Fig. 13 a, b and c for *ED* and in Fig. 13 d, e and f for *MVD*. The observations when the FSSP did not operate properly due to installation of the brake were excluded from the inter-comparison. The agreement for both sizing parameters was found good ($R^2$ = 0.80, slope 0.79 and $R^2$ = 0.78, slope 0.75 respectively). The best agreement was observed when the wind direction (see Fig.13a, d) was inside the range of iso-axial conditions where all the points were focused along the 1:1 line. When the direction was perpendicular the points were spread wider (maximum observed difference between the





two probes was about 20%). Surprisingly, despite the fact that CAS was measuring lower $N_c$ even by a factor of 10 when the wind direction was perpendicular to CAS fixed direction, the derived *ED* and *MVD* were not significantly influenced. Both sizing parameters were derived from the measured size distribution as described in section 2.3. We found that even if significant number of cloud droplets was lost due to inertia, the shape and the position of the peak of the size distribution measured by

CAS remained similar. This behaviour was found the same through the whole available cleaned data set (~ 183 hours) with the maximum *ED* and *MVD* of 35 and 30 μm. It has to be pointed out that this behaviour might be exclusive for sub-Arctic conditions with generally small cloud droplets. This fact allows us to use the majority of the data set when investigating those two derived sizing parameters. As a result, it creates a significant and usable data set without need to disqualify data according to particular wind direction. Thus we obtain statistically significant size properties of the cloud droplets at wide range of

meteorological conditions. We also investigated the probability that wind speed will affect the sizing parameters (see Fig. 13c, f). When the probes were facing high wind speed, *ED* and *MVD* were slightly influenced (FSSP derived bigger values of *ED* and *MVD* when compared to CAS). On the other hand, while they were facing low wind speeds sizing were again influenced on the opposite way (FSSP derived smaller values when compared to CAS). This could happen due to the isokinetic motion of the particles. The larger particles could not enter the FSSP because the inner diameter necking on the inlet was changing

from 3.8 to 2.0 cm. Finally, Fig. 13 b, d indicates that at lower temperatures we observed smaller *ED* and *MVD* values.

According to previous analysis, our main conclusion was that even if there were slight biases and uncertainties the agreement in inter-comparison was considered good as both $R^2$ and slope were higher than 0.75. As a result, we propose when deriving the sizing parameters, *ED* and *MVD*, all measurements can be used for further research after carefully exclude the FSSP dataset that was obtained from the wind rose sector where the brake influenced its performance.

**3.5  Evaluation of the CDP during PaCE 2013.**

After comparing and analyzing discrepancies and biases between the CAS and the FSSP cloud probes, we discuss the performance of the CDP cloud probe separately. To evaluate CDP performance we used only FSSP data. We should remind that during the period that both the CDP and the CAS probe were on site (from 15 October to 27 November), the CDP encountered several malfunction and operational problems during icing conditions. As a result, there was a lack of common

data between the CDP and the CAS probe.

We used CDP and FSSP data from September 25th to October 14th (~ 70 hours of cleaned data set) since it was the only period that the CDP faced fewer operational problems since average ambient temperatures were mostly above 0 °C. During this inter-comparison, a set of data from the FSSP was removed (0 to 74°, where the FSSP had significant malfunctions due to installation of the brake). For this time period, average temperature at the station was 1.7 (std 1.6) °C and the averaged wind

speed was 6.9 (std 3.6) ms$^{-1}$. In order to compare similar size range for both cloud probes the first 2 bins from the FSSP were removed. This means that the following results depict size range from 3.5 to 47 μm for the FSSP and from 3 to 50 μm for the CDP probe.





As it was already mentioned at the beginning of section 3.2 those two instruments belong to the sub category of the probes that were installed on a rotating platform during PaCE 2013. Figure 14 shows, as it was expected, that the ability of the two instruments to derive $N_c$ was good ($R^2 = 0.84$, slope 1.11). However, there were cases where the difference between them was about 30%. Additionally, we investigated the derived sizing parameters *ED* and *MVD*, see fig.15. In a range of temperatures (from -3.9 to 3.8 °C) and wind speeds (0.9 to 19 ms$^{-1}$) agreement corresponding to the sizing parameters was good ($R^2 = 0.82$ and 0.79 with slopes 1.23 and 1.25 for *ED* and *MVD* respectively). However, when FSSP derived *ED* and *MVD* for sizes larger than 22.5 µm, we could see a difference that could be even 15 µm smaller in comparison with CDP. This difference was noticed especially when the wind speeds were low. FSSP had similar behavior (section 3.4) when we were comparing CAS and FSSP due to isokinetic motion of the particles.

A significant limitation in derived *LWC* regarding temperature was already discussed above during the comparison of CAS and FSSP. In this case, the temperature ranged from -3.9 to 3.8 °C. This range that was above -4 °C (the temperature point that was set in section 3.4). As a result, the comparison of CDP with FSSP derived *LWC* did not lead to reasonable correlation and no *LWC* data are presented here.

In summary, the CDP was operating well in warm liquid clouds and had good agreement in cloud droplet counts and the sizing derived parameters with FSSP. On the other hand, while we faced sub-zero conditions the CDP operation was considered as problematic. Its probe inlet became often clogged because of supercooled cloud drops accumulation. This happened to both the rotation and inhalation system because its big surfaces were collecting ice and it had a small opening of inhalation system. In conclusion, even that this CDP setup performed well in warm cloud conditions, it is not suitable instrument for semi-long term ground based measurement of clouds in sub-Artic conditions, when we are facing subzero conditions.

## 4. Conclusions

We conducted ground based in-situ cloud measurements during PaCE 2013 from September 14[th] until November 28[th]. We deployed three cloud spectrometers (CAS, FSSP and CDP) on the roof of Sammaltunturi station, located in Finnish sub-Arctic. The obtained data set was analyzed in detail to evaluate the instruments ground based setups' performance and established limitations for future studies. All cloud spectrometers and their setups are owned by FMI and results could be used in campaigns with similar meteorological conditions, sub-Arctic conditions with frequently occurring supercooled clouds. The CAS was installed and fixed against the main wind direction of the station (~ 225°) and the other two probes (FSSP and CDP) were installed on rotating platforms. Each probe had its own inhalation system. Their ability to measure the size distribution of cloud droplets along with the derived number concentration ($N_c$), the sizing parameters (*ED* and *MVD*) and the liquid water content (*LWC*) was tested and the above parameters were mutually compared. In this work, CAS and FSSP ground setups were investigated first because their parallel data coverage was the best (~ 243 hours of common cleaned data set). On the other hand, CDP had low common data set with the CAS. The reason for that was that from the date that the CAS was installed we were mainly facing sub-zero temperatures, conditions that proved that were not favorable for this CDP ground setup.





Regarding the size distribution, we noticed some differences in our measurements. Even though all three probes were calibrated the same way, but each separately, we found that their sizing was slightly different in real atmospheric conditions. There was a slight shift in FSSP sizing towards bigger sizes in comparison to the CAS probe, ~1.5 μm in size range from 7 to 10 μm and a slight shift in CDP sizing towards smaller sizes in comparison with the CAS probe ~2.5 μm in size range from 5 to 7.5 μm. Our conclusions on the four derived parameters should take into account those sizing uncertainties. The FSSP, an instrument placed on rotational platform, with wider inlet opening of inhalation system, provided the best performance and data coverage for in-situ cloud droplets measurements. The CDP probe often accumulated ice in sub-zero condition, both in its rotational platform and inhalation system. This was happening due to presence of supercooled clouds at the station. The big surfaces of the CDPs rotation platform and the inlet with small opening were collecting ice very fast. However, when the station was in warm cloud and the temperature was above zero, CDP was operating well considering the cloud droplets counting.

To estimate the droplet counting performance and possible droplets losses, we used number concentration ($N_c$). Results indicated that when we were deriving $N_c$, the mutual direction of probe heading and the wind direction were playing the most significant role. From the inter-comparison of the CAS (fixed orientation) against FSSP (rotating platform), it was found that the CAS probe had the best agreement ($R^2$=0.70) with the FSSP during wind iso-axial conditions (200 to 235°). The CAS probe counting efficiency was strongly dependent on the wind direction, this can be clearly explained by its installation to fixed orientation. When the station was inside warm clouds, both the CDP and the FSSP had good agreement ($R^2$= 0.82) as they were both operating on rotating platforms.

The *LWC* was found the most sensitive derived parameter. This is because *LWC* strong dependency both on size and the number of droplets in each size bin. Thus, the wind direction played again the most significant role. For that reason, we strongly recommend that CAS and FSSP derived *LWC* values only from wind iso-axial conditions should be used. Additionally, *LWC* values were found also temperature dependent. For temperatures lower than -4 °C the agreement between the CAS and the FSSP in wind iso-axial conditions was high ($R^2$ =0.62) and that is why temperature -4 °C was adopted as the critical temperature point regarding *LWC* estimation. We excluded all *LWC* values derived from the CDP due to its problematic operation at sub-zero conditions i.e. close to temperature -4 °C , CDP was usually frozen.

The analysis of the derived sizing parameters, *ED* and *MVD*, showed good agreement among the three probes during the time they were operating properly. However, our conclusions here concentrate only to CAS and FSSP, because only those two instruments were operating properly in subzero temperatures, temperatures that we usually face during PaCE. The obtained inter-comparison results were surprisingly good even though CAS lost a significant amount of cloud droplets due to its orientation. The wind direction did not significantly affect neither the *ED* values nor the *MVD*, even though large discrepancies (uncertainty ~ 85%) in $N_c$ outside of the wind iso-axial conditions could be found (e.g. when the wind direction was perpendicular to the CAS probe fixed direction, uncertainty for sizing parameters was ~ 20%). The *ED* and *MVD* was not affected because the shape and the peak position of the CAS size droplet distribution did not change significantly. Such behavior held through ~ 183 hours of data set. This result is important as it allows us to use a larger data set without limitations



due to wind direction and other meteorological parameters regarding derived *ED* and *MVD*. The small differences (about 2 μm) were explained by a closer look in size distribution of each spectrometer and the differences in sizing during operation in real conditions as they were mentioned above.

5    Our final recommendations and our view on the main limitations of each spectrometer ground setup for using and analyzing the obtained data sets during sub-Arctic meteorological conditions with frequently occurring supercooled clouds (including future PaCE campaigns) are summarized and presented in Table 2.



**Data availability**

The cloud probes and meteorological data used here are available upon request to the corresponding author (Konstantinos.doulgeris@fmi.fi).

**Author contribution**

KMD, DB wrote the manuscript with contribution from all co-authors. KMD prepared the manuscript and analyzed data from all cloud probes and meteorological data with contribution from DB. DB installed and operated all the instruments during PACE2013. MK and SR provided the CDP ground set up and performed its calibration.

**Competing interests**

The authors declare that they have no conflict of interest.

**Acknowledgments**

This work was supported by KONE foundation, Nordforsk Contract number 26060, CRAICC Amendment on CRAICC-PEEX Collaboration, Academy of Finland project: Greenhouse gas, aerosol and albedo variations in the changing Arctic (project number 269095), Academy of Finland Center of Excellence program (project number 307331), BACCHUS (EU 7th Framework program), Natural Environment Research Council (NERC), grant number NE-L011514-1 and ACTRIS-2, the European Research Infrastructure for the observation of Aerosol, Clouds, and Trace gases. This project has received funding from the European Union's Horizon 2020 research and innovation programme under grant agreement No 654109.

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





## Figures

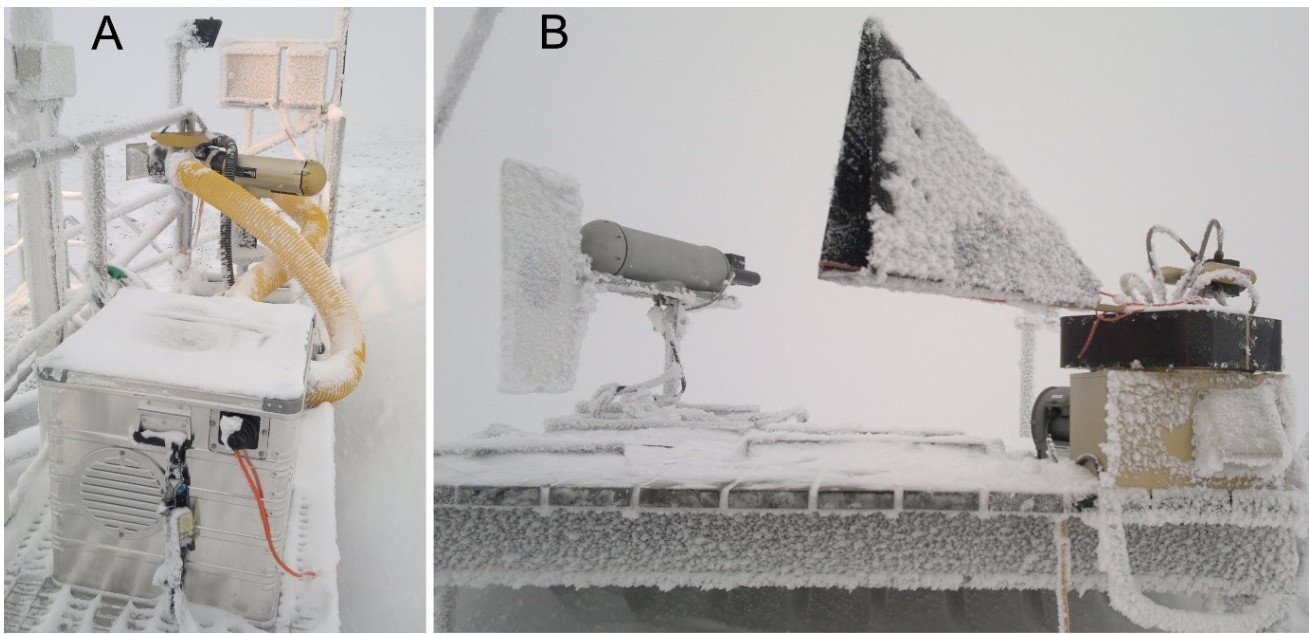

Figure 1. A) The CAPS probe setup and B) the FSSP-100 and the CDP setups as they were installed on Sammaltunturi station during PACE2013.

Table 1. Instruments, wavelengths (nm), sampling area (mm), number of bins, probe air speed (ms$^{-1}$), size range (μm), time resolution (s), operation starting and ending date are presented.

| Instrument | Laser Wavelength (nm) | Sampling area (mm$^2$) | Number of bins | Probe air speed (m/s) | Size range (μm) | Time resolution (s) | Operation start date | Operation end date |
|---|---|---|---|---|---|---|---|---|
| CAPS | 680 | 0.24 | 30 | 17-23 | 0.61 - 50 | 1 | 15 Oct | 28 Nov |
| FSSP | 633 | 0.414 | 40 | 10 | 1.2 - 47 | 1 | 14 Sept | 28 Nov |
| CDP | 658 | 0.3 | 30 | 14 | 3 - 50 | 1 | 25 Sept | 28 Nov |



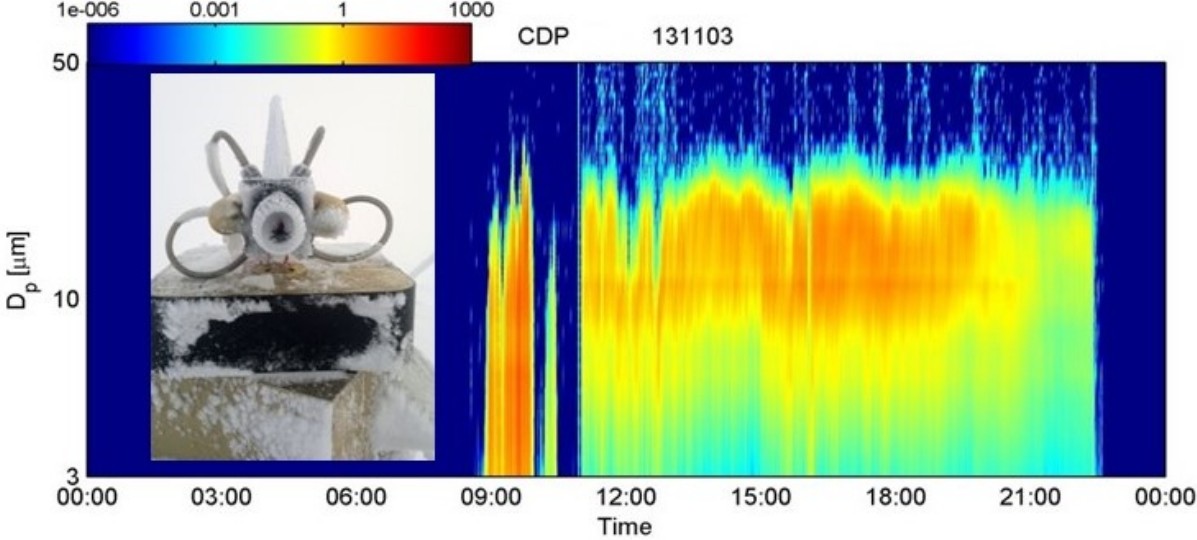

Figure 2. An example case of the CDP probe where its rotational platform and inlet are frozen. The size distribution of the CDP probe at 03.11.2013 is depicted. The instrument was out of order from 00:00 to about 11:30 and CDP cleaning procedure was needed to start operation again.





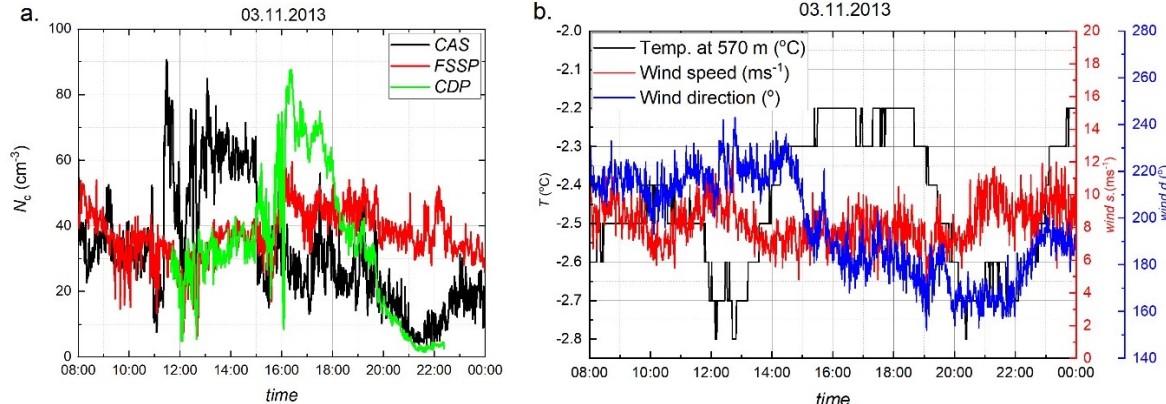

Figure 3. Time series of 1 minute averages during 03.11.2013; (a)$N_c$ of each cloud probe along with (b) temperature, wind speed and wind direction are presented. This is a typical example of the cloud probes accumulating ice. From 12:00 we can see drop in $N_c$ of the CAS. The sudden increase just before was a clear sign of probes inlet freezing. The same behavior could be also seen for the CDP after 16:00. When ice was accumulated, the opening of the probe inlet was getting smaller which resulted in a raising PAS. During deriving $N_c$ to evaluate our data set, we considered the PAS was constant. The underestimation of the PAS explains the abnormality in $N_c$.

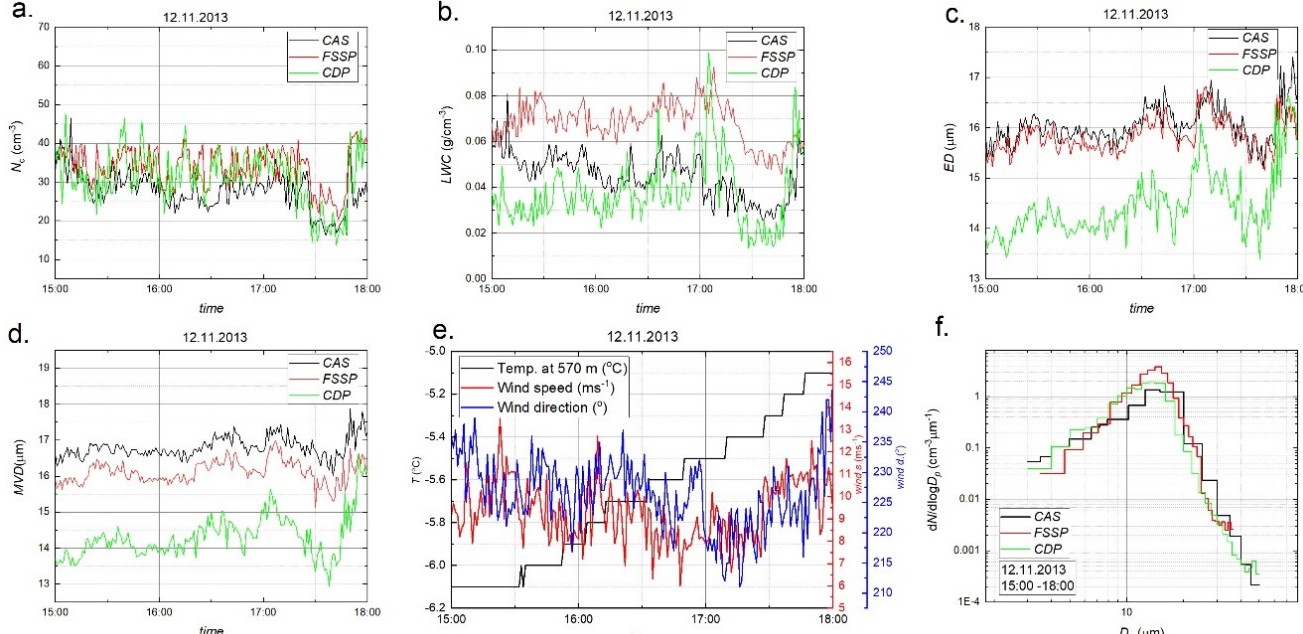

Figure 4. Time series on 12 November case from 15:00 to 18:00, the main parameters as they were derived /measured from all three cloud probes: (a) $N_c$; (b) $LWC$; (c) $ED$; (d) $MVD$; (e) temperature, wind speed, wind direction and (f) size distribution. All three instruments were pointing to the same direction. This resulted high agreement in $N_c$ for all three probes. In addition, we also noticed good agreement in $LWC$. The main reason for slight differences in $ED$ and $MVD$ was the different sizing ability of the probes.





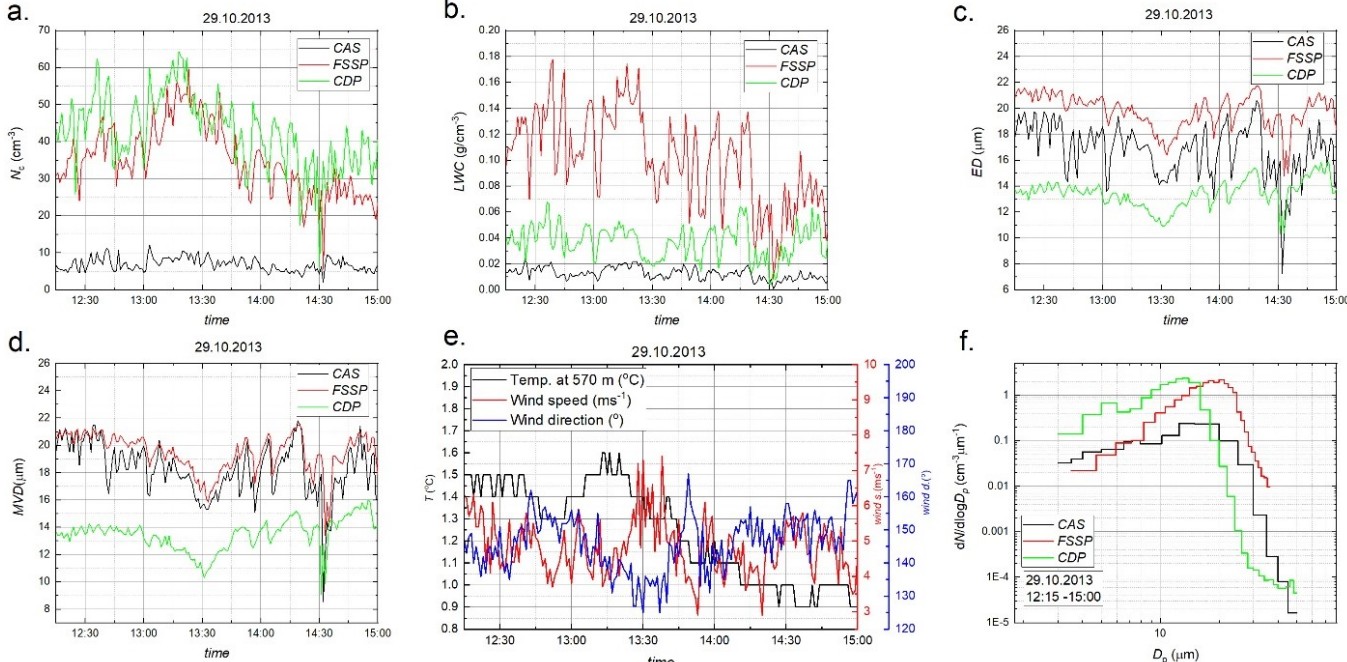

Figure 5. Time series on 29 October from 12:15 to 15:00, the main parameters as they were derived /measured from all three cloud probes: (a) $N_c$; (b) $LWC$; (c) $ED$; (d) $MVD$; (e) temperature, wind speed, wind direction and (f) size distribution. The wind direction was perpendicular to the CAS probe. This resulted in CAS significantly underestimated $N_c$ and $LWC$. The main reason for slight differences in $ED$ and $MVD$ was the different sizing ability of the probes.

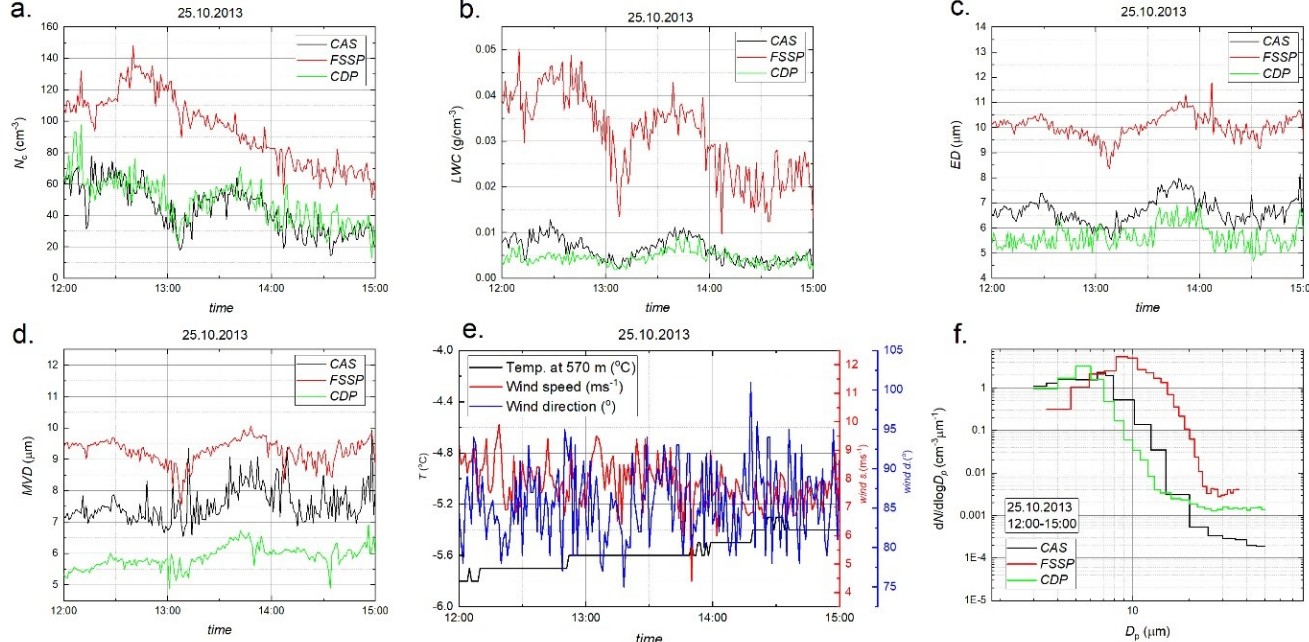

Figure 6. Time series on 25 October from 12:00 to 15:00, the main parameters as they were derived /measured from all three cloud probes: (a) $N_c$; (b) $LWC$; (c) $ED$; (d) $MVD$; (e) temperature, wind speed, wind direction and (f) size distribution. During this case the probes were not facing the same direction. Agreement in $N_c$ between CAS and CDP indicates that CDP rotation system was frozen and CAS not facing the wind. Large discrepancies were observed in $LWC$ and slight discrepancies in $ED$ and $MVD$. The main reason for those discrepancies was the different sizing ability of the probes.



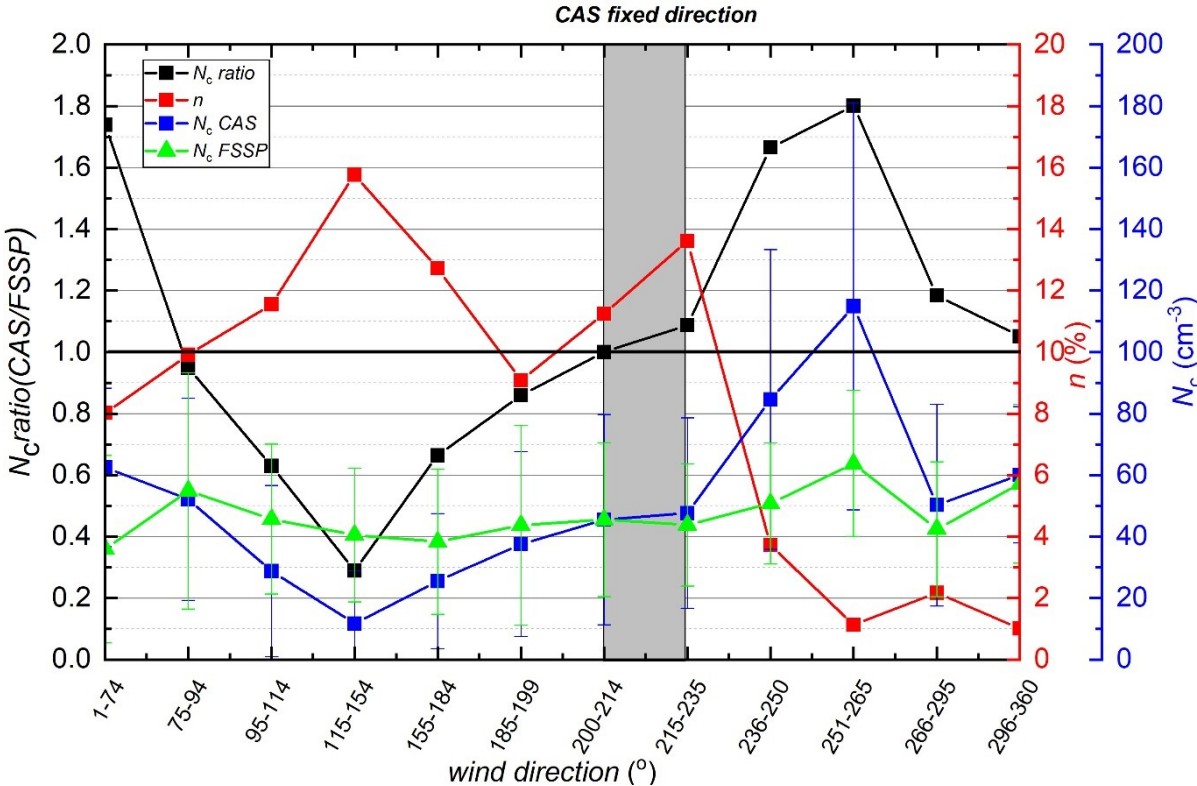

Figure 7. Number concentration ratio ($N_{c\ ratio}$), number of observations (n), and $N_c$ of the CAS probe and the FSSP for each part of the 360° wind rose as it was divided for detailed investigation. The grey rectangle corresponds to wind iso-axial measurements.







Figure 8. Comparison of number concentration ($N_c$) as it was derived from the CAS and the FSSP is presented for two sectors of the wind rose during the station was inside a cloud (a) 200 to 214°; (b) 215 to 235°; Color code represent the wind direction, (c) and (d) the size distribution as it was measured from the CAS probe and the FSSP for the same two wind sectors is presented.





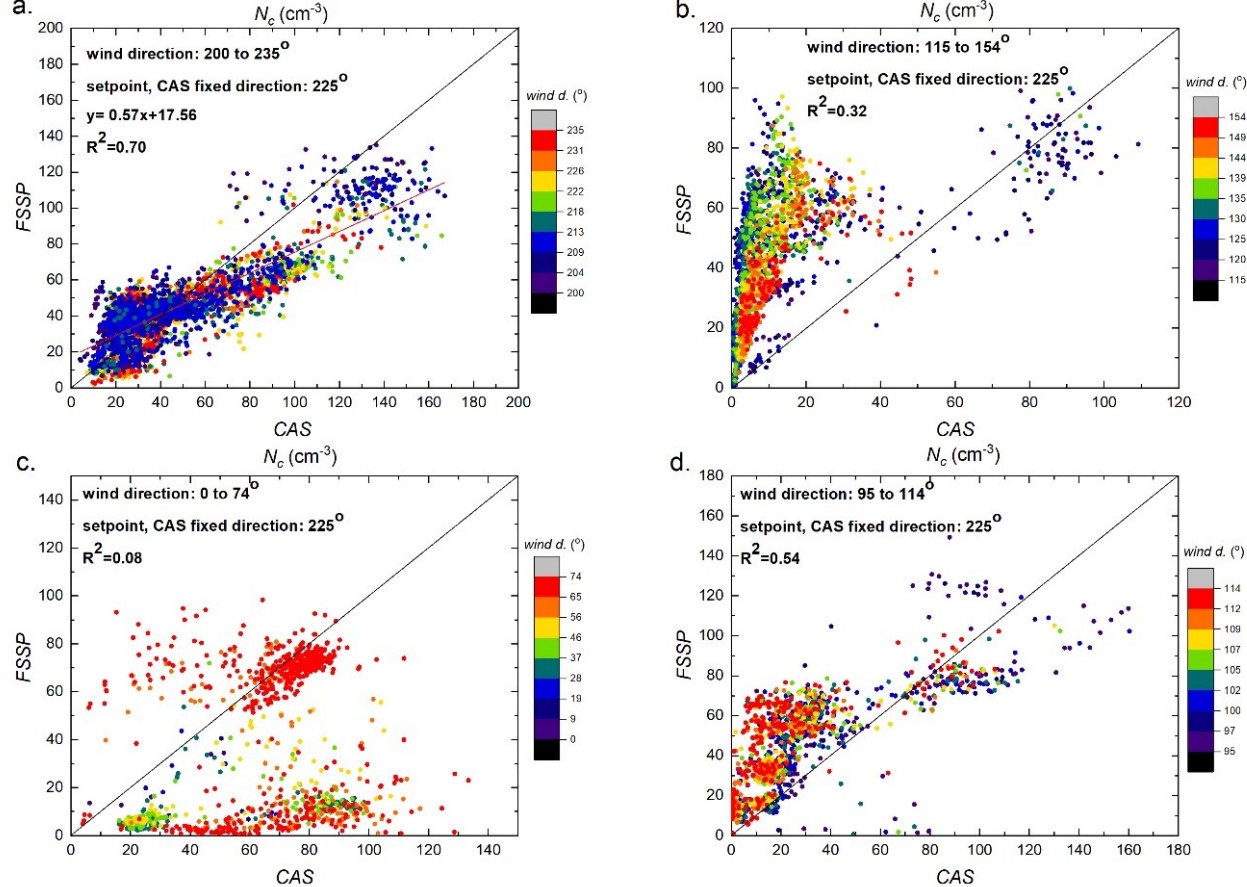

Figure 9. Comparison of number concentration ($N_c$) as it was derived from the CAS and the FSSP is presented for four different representative sectors of the wind rose during the station was inside a cloud. (a) 200 to 235° represented observations during the wind iso-axial conditions; (b) 115 to 154° represented observations during wind direction was perpendicular to the fixed CAS direction; (c) wind sector (0 to 74°) where the FSSP had operation problems due to its brake installation and (d) one wind sector when the wind direction was between iso-axial and perpendicular conditions (95 to 114°).



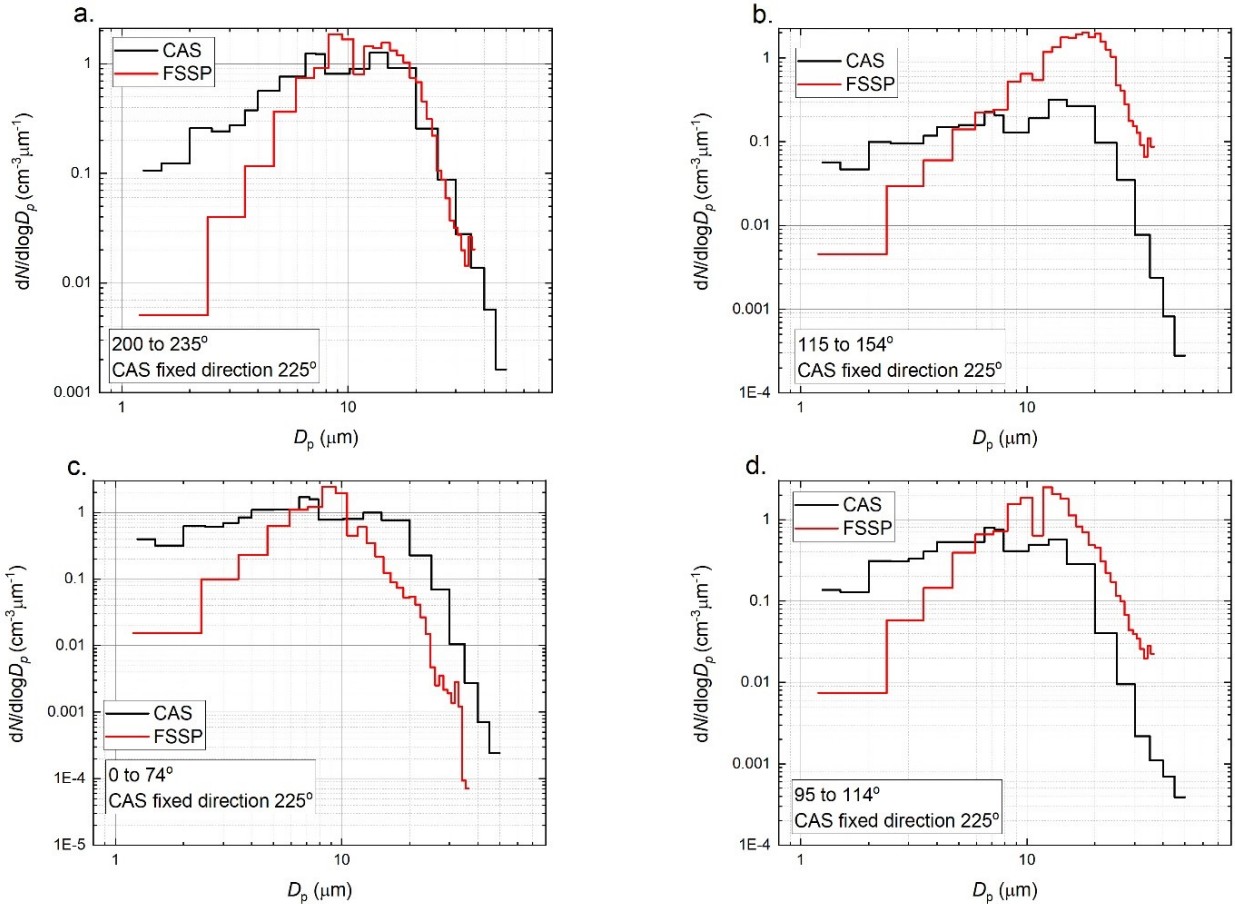

Figure 10. Size distributions of the CAS and the FSSP for four different representative sectors (same to Fig. 9) of the wind rose (a) 200 to 235° represented observations during the wind iso-axial conditions; (b) 115 to 154° represented observations during perpendicular wind direction to the fixed CAS direction; (c) wind sector (0 to 74°) where the FSSP had operation problems due to its brake installation and (d) one wind sector when the wind direction was between iso-axial and perpendicular conditions (95 to 114°).





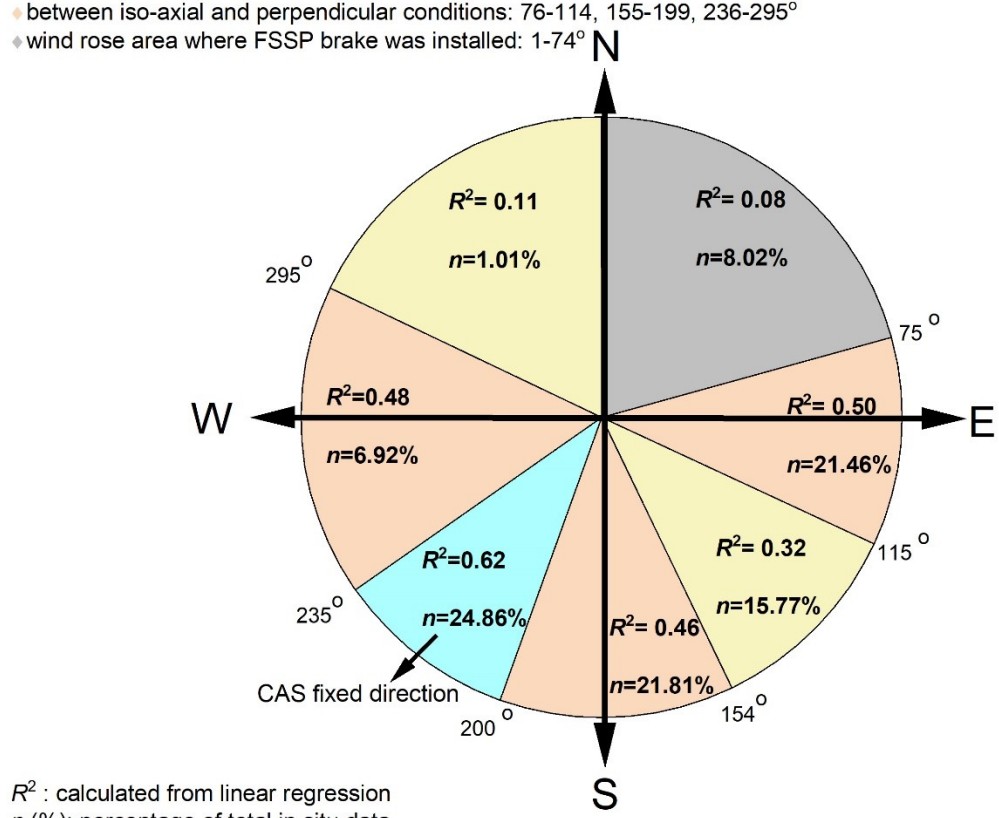

Figure 11. Inter-comparison of number concentration ($N_c$) between the CAS and the FSSP based on the wind direction. The CAS was set and installed in a fixed direction (southwest, ~ 225°); the FSSP was installed on a rotating platform and following the wind direction. The wind rose was separated to four representative wind direction conditions; wind iso-axial conditions (from 200 to 235°), perpendicular conditions (from 115 to 154 and 296 to 360°), conditions between iso-axial and perpendicular (from 76 to 114, from 155 to 199 and from 236 to 295°) and the special case where the brake influenced the performance of FSSP (from 1 to 74°).





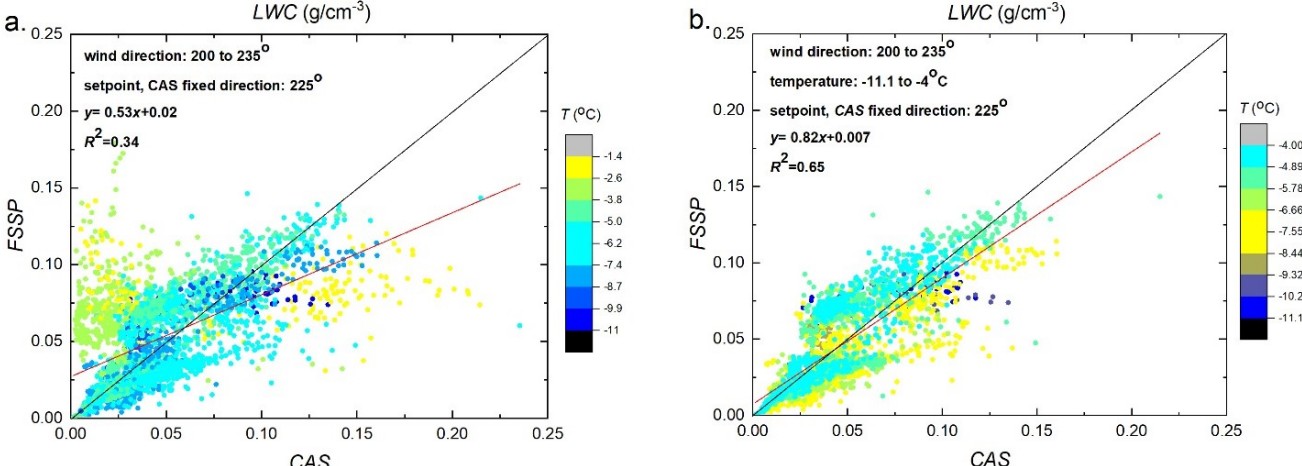

Figure 12. (a) Comparison of *LWC* as it was derived from the CAS and FSSP for wind iso-axial conditions, color code represents full temperature range from-11.1 to -1.4 ℃ (b) Comparison of *LWC* as it was derived from the CAS and FSSP is presented for the same conditions but only for temperature range from -11.1 to -4 ℃.





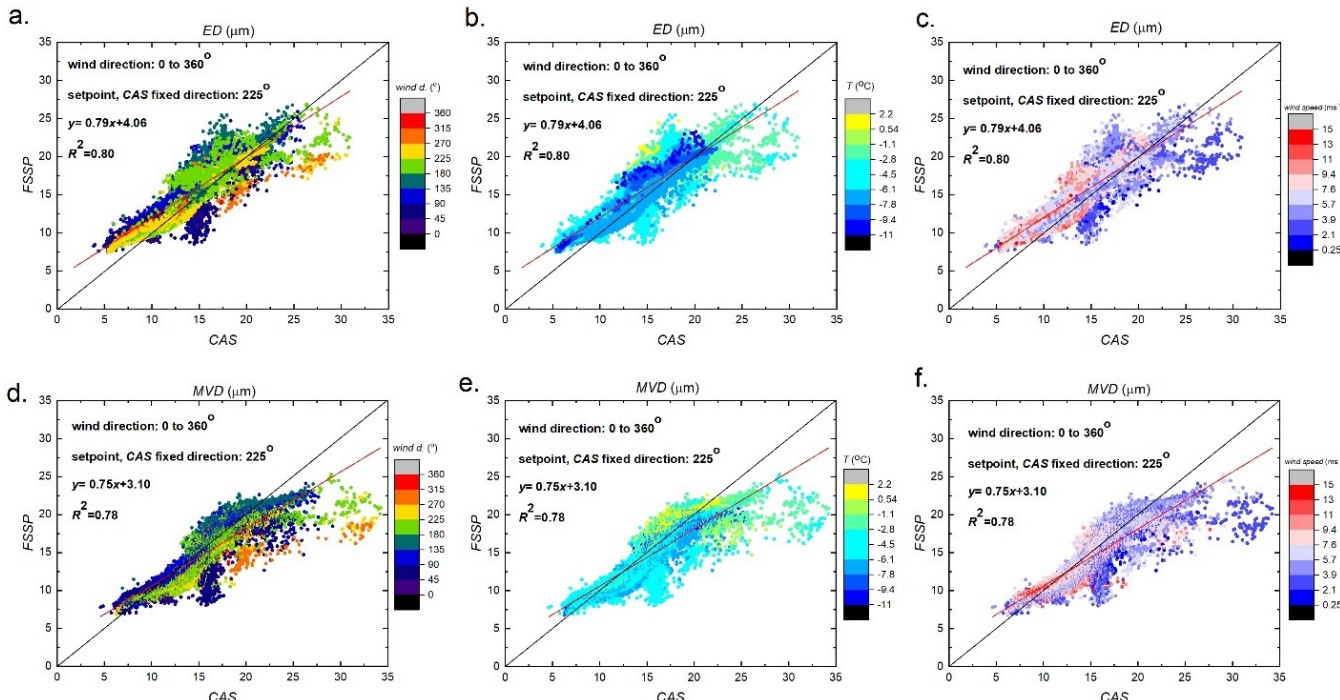

Figure 13. Comparison of *ED* (a) (b) (c) and *MVD* (d) (e) (f) as it was derived from the CAS and FSSP is presented for all wind directions. Color code represents (a) and (d) wind direction; (b) and (e) temperature; (c) and (f) wind speed.

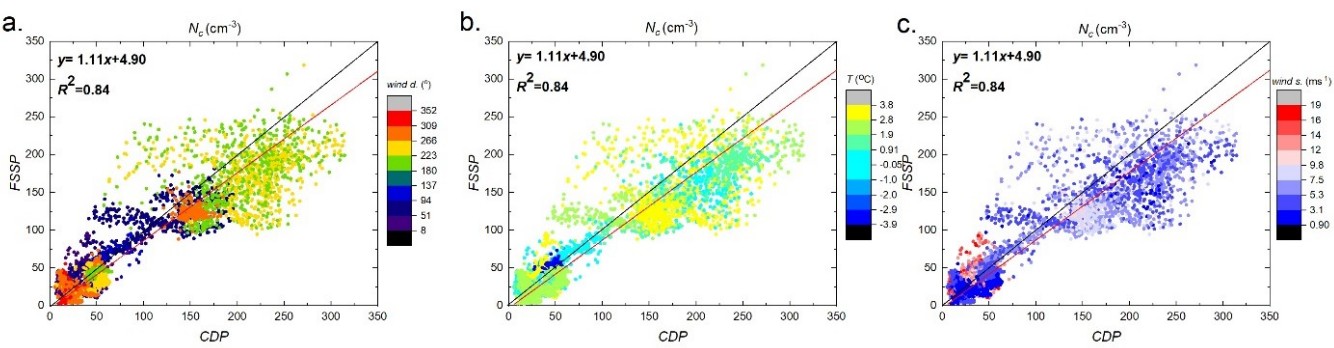

Figure 14. Inter-comparison of number concentration ($N_c$) as it was derived from the CDP and the FSSP is presented for all wind directions. Color code represents (a) wind direction; (b) temperature and (c) wind speed.



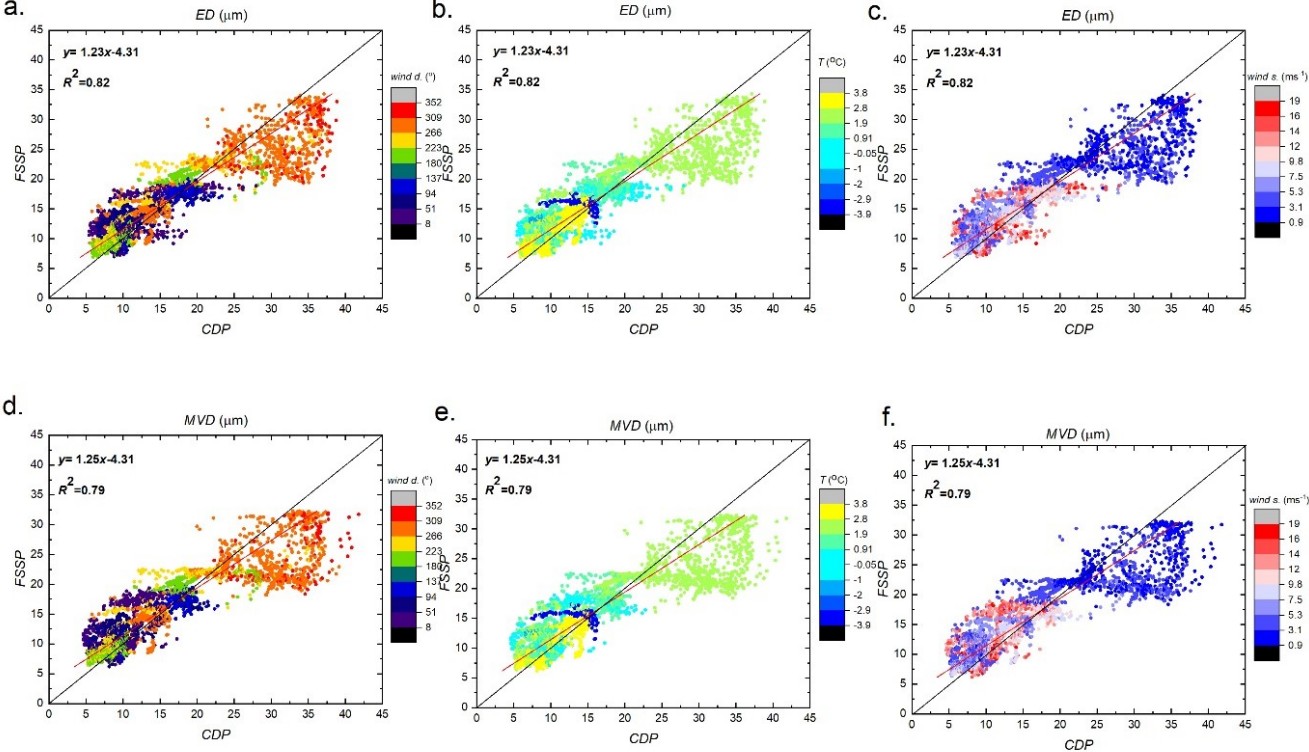

Figure 15. Comparison of *ED* (a) (b) (c) and *MVD* (d) (e) (f) as it was derived from the CDP and FSSP is presented for all wind directions. Color code represents (a) and (d) wind direction; (b) and (e) temperature and (c) and (f) wind speed.



Table 2 presents the final recommendations for data analysis regarding the cloud spectrometers ground based setups for future campaigns in sub-arctic conditions with frequently occurring supercooled clouds.

| | $N_c$ | *ED, MVD* | *LWC* | *Comments* |
|---|---|---|---|---|
| *CAS* | Only data from wind iso-axial conditions. (± 20° from its fixed direction) should be used | All measurements can be used for further analysis, independent on wind direction in size range of ED and MVD 5 - 35 μm | Only data from wind iso-axial conditions and temperatures below – 4 °C should be used | Good data coverage (67%), Operating properly both in non-icing and icing conditions, needs daily cleaning |
| *FSSP* | Data from all wind sectors will be used except data from wind sector where brake was installed (± 40° brake direction) | All data can be used for further analysis except data from wind sector where brake was installed (± 40° from brake direction) | Only data from wind iso-axial conditions and temperatures below – 4 °C should be used | The best data coverage (75%), Operating properly both in non-icing and icing conditions, needs daily cleaning |
| *CDP* | Usable in warm clouds. Limitation in temperature, operational problems at sub-zero temperatures | All data obtained in non-icing conditions can be used for further analysis | Not usable due to temperature range. | Low data coverage (17%), Operating properly in non-icing conditions, not recommended for sub-zero temperatures |