# Peer review of "In-situ cloud ground based measurements in Finnish sub-Arctic: Intercomparison of three cloud spectrometers setups."

_Atmospheric Measurement Techniques, 2020_

## Referee Comment (RC1) · Darrel Baumgardner (Referee) · 16 Mar 2020

The study that is discussed in this submission focuses on the performance of three cloud probes that were originally designed for operation on aircraft but an attempt has been made to adapt then to a ground based location where the environment can often be quite harsh with respect to icing conditions. I think that given the multiple issues with how the instruments were operated, this paper is overly lengthy. It could just as easily have been a very short technical note that points out how you should NOT mount and operate instruments on the ground that are designed to be used for aircraft. What is puzzling is why the decision was made to use these instruments rather than the DMT

[Figure]

Fog Monitor that utilizes the same measurement theory but was intentionally designed for ground based measurements. They do not explain why they chose to mount and operate the instruments in the way they did instead of using a wind tunnel set up that would have circumvented the problems that arose.

As a three page note, there would be two principal conclusions: 1) Cloud probes with inlets should always be mounted into the wind and 2) proper deicing is always necessary when conditions dictate it.

As a study comparing three instruments with similar measurement techniques it is not as useful or relevant as the number of other studies where instruments are compared in wind tunnels, such as the study done at Puy-de-Dôme referenced in the current study.

The current study would be much more useful and relevant if it encompassed not only the measurements at their site but discussed why they chose to operate their instruments as they did compared to other sites such as Storm Peak, Elk Mountain, Puy-de-Dôme, Jungfraujoch, and the Zugspitze where similar studies have been done but more successfully. Storm Peak, Elk Mountain and Puy-de-Dôme all use a wind tunnel to introduce cloud air to the instruments so that the sensors are being under conditions more like they were designed for, i.e. aircraft.

I can't recommend this manuscript for publication in its present form as I don't find the results that useful other than as a warning about how not to operate these instruments. A more comprehensive review of ground based measurements with sensors designed for aircraft would be far more useful.

Although I am the chief scientist and founder of Droplet Measurement Technologies, I was not involved with the setting up of the instruments that were involved in this study or the ventilation systems used to introduce cloud air. I have tried to ascertain how this all evolved but the technical staff who were involved are no longer with the company so I have no way of understanding the history of this project. I would recommend that the authors consider a different approach for future studies.

---

## Referee Comment (RC2) · Anonymous Referee #3 · 28 May 2020

The submitted manuscript serves as a record of activities involving three cloud spectrometers during the Pallas Cloud Experiment in 2013. Whilst the title suggests it is an intercomparison between these instruments, it is really a comparison of the rather specific experimental setups employed during this campaign. The paper therefore serves more as a campaign report than a reference on how best to set up experiments for ground-based in-situ cloud measurement. Nevertheless, it does highlight some of the pitfalls.

I am aware of the motivation within communities such as ACTRIS for the establishment of long-term ground-based in-situ cloud measurements, and as such this paper is a

step in the right direction with respect to evaluating how these might be established. However, the conclusions do not seem robust enough to form the basis of wider recommendations. The paper does highlight the considerable difficulties faced by any attempt at long-term observations, and it is evident that any plans for unattended operation would pose particular challenges, especially in the sub-Arctic environment. In fairness, the authors are conservative in their recommendations and focus on Pallas campaigns (past analysis and future experiments).

I note that this experiment was performed contemporaneously with that at Puy-de-Dôme (Guyot et al, 2015) and hence the insight from the findings of the latter were not available to provide guidance on what complementary instruments, based on ensembles of particles, might be installed and used to explore scaling of the number concentration and related parameters. In fairness, the authors do recommend such instrumentation for future campaigns.

Guyot et al (2015) found that FSSP measurements suggested anisokinetic sampling and a high sensitivity to the wind speed and direction. It would be helpful if the authors could comment on how their findings relate to this earlier analysis.

The authors go into great detail regarding alignment relative to the wind direction, and the discussion is rather laboured and lengthy. The effects on number concentration are not particularly surprising, but are elaborated in great detail, no doubt because the specific instrumental setups (e.g. the brake on the FSSP) require it. This discussion may benefit from being shortened.

I note the authors specifically mention the frequent occurrence of supercooled clouds at this location. Do they have further corroborative evidence that the clouds being sampled contained only supercooled liquid water drops. Whilst LWC is readily calculated in terms of the measured parameters, it would be useful if the authors could comment on whether any data relate ice particles.

Whilst the manuscript appears to be in scope for the journal, I would recommend revi-

sion before it could be considered for publication.

On a technical level, I believe the quantity $pro_i$ defined on p.7 line 22 should be the reciprocal of that displayed. Also, the quantities $b_{i+1}$ should, I believe, be $b_{i^*+1}$.
* * *

---

## Author Comment (AC2) · 25 Jun 2020

We sincerely thank the reviewer for carefully reading of our work, for his review and valuable comments. We have carefully reviewed the comments and have revised the manuscript accordingly. Our responses are given in a point-by-point manner below.

Reviewer comment (RC)

Authors answer (AA)

RC1:

*Whilst the title suggests it is an intercomparison between these instruments, it is really a comparison of the rather specific experimental setups employed during this campaign. The paper therefore serves more as a campaign report than a reference on how best to set up experiments for ground-based in-situ cloud measurement. Nevertheless, it does highlight some of the pitfalls.*

AA1:

We agree with the reviewer that this was an intercomparison between the specific ground setups and not the instruments themselves and we clarified this through the manuscript (e.g. p1, line 15, "*The main motivation of the campaign was to conduct in-situ cloud measurements with three different cloud spectrometer probes and perform an evaluation of their ground based setups*"). We consider that this work was not just an intercomparison but also an operative experiment on how to operate cloud probes for ground based measurements during harsh conditions. We will change the manuscript title to avoid any misunderstandings and possible confusions to: "*In-situ cloud ground based measurements in Finnish sub-Arctic: Intercomparison of three cloud spectrometers setups* ". Also, we clarified in the abstract of the revised manuscript that we intercompared the experimental ground setups of the cloud probes.

p1, line 18: "We investigated how different meteorological parameters affect each instrument operation" to "We investigated how different meteorological parameters affect each instruments' ground based setup operation"

p1, line 20: "we suggested limitations for further use of the instruments in campaigns where focus is on investigating aerosol cloud interactions" to "we suggested limitations for further use of the instruments setups in campaigns where focus is on investigating aerosol cloud interactions"

p1, line 24: "A complete intercomparison between the CAS probe and the FSSP-100 and additionally between the FSSP-100 and the CDP probe was made and presented." to "A complete intercomparison between the CAS and the FSSP-100 ground setups and additionally between the FSSP-100 and the CDP ground setups was made and presented".

RC2:

*I am aware of the motivation within communities such as ACTRIS for the establishment of long-term ground-based in-situ cloud measurements, and as such this paper is a step in the right direction with respect to evaluating how these might be established. However, the conclusions do not seem robust enough to form the basis of wider recommendations. The paper does highlight the considerable difficulties faced by any attempt at long-term observations, and it is evident that any plans for unattended operation would pose particular challenges, especially in the sub-Arctic environment. In fairness, the authors are conservative in their recommendations and focus on Pallas campaigns (past analysis and future experiments).*

AA2:

We thank the reviewer acknowledging the demand within community for long-term ground-based in-situ cloud measurements and understanding the main motivation of our work. We indeed are conservative and focus on Pallas campaigns. Our recommendations are based on results we obtained from continuous (about two and a half months) PaCE campaign at harsh sub-Arctic conditions. There are two main conclusions that we highlight as basis for wider recommendations. Conclusions were modified accordingly:

P17, line 4: "…were mentioned above. As final suggestions regarding performing continuous ground based in-situ cloud measurements in harsh environments, we would like to highlight two major issues. First, the cloud probes should always continuously face the wind direction to minimize the sampling losses. If this is not secured, only the measurements that were conducted in wind iso-axial conditions can be used for further analysis. However, deriving the sizing parameters *ED* and *MVD* for the whole wind direction spectrum is still possible, but must be done with insight and prudence. Secondly, the cloud probes need necessary daily or more frequent checkups and cleaning of their inlets."

RC3:

*I note that this experiment was performed contemporaneously with that at Puy-de Dôme (Guyot et al, 2015) and hence the insight from the findings of the latter were not available to provide guidance on what complementary instruments, based on ensembles of particles, might be installed and used to explore scaling of the number concentration and related parameters. In fairness, the authors do recommend such instrumentation for future campaigns.*

AA3: We agree with the reviewer and we indeed recommend that such instrumentation can be really useful for future or similar campaigns, especially for measuring *LWC* (p13, line 22 *". In addition, we suggest the deployment of another LWC sensor, e.g. the particle volume monitor (PVM-100, Gerber 1999) during future campaigns in order to obtain another reference LWC values for inter-comparison in wide temperature range"*).

In the revised manuscript the following will be added after previous sentence

p13, line 23: "… temperature range. The current market does not offer an instrumentation fulfilling our requirements. However, we are continuously following the development of a new generation of counters designed for ground based in-situ cloud measurements. Thus, it is a matter of future deployment during upcoming PaCE campaigns. "

RC4:

*Guyot et al (2015) found that FSSP measurements suggested anisokinetic sampling and a high sensitivity to the wind speed and direction. It would be helpful if the authors could comment on how their findings relate to this earlier analysis.*

AA4:

According to Guyot et.al. (2015) for wind speeds larger than 3 m/s, the sensitivity of FSSP to the wind direction was high (wind speed average value in Pallas was 6.8 m/s) (page4360 in Guyot et al (2015) *"On average, the greater the angular deviation from isoaxial configuration is, the more the size distribution is reduced, except for a 3 m/s wind speed"*). We agree that this was the main reason that caused discrepancies to the fixed direction of CAPS cloud probe ground setup (the only instrument with fixed direction, discussed in detail (section 3.3)) and it mainly affected its size distribution and hence the number concentration. In our case FSSP was continuously following the wind direction without the need of man force in comparison with Puy-de-Dôme. For this reason, FSSP sensitivity was mainly connected with its brake installation and not the anisokinetic sampling. We should also highlight that in comparison to Guyot et.al. (2015) where they conducted measurements in temperatures above zero, we were usually facing temperature below zero. In revised manuscript section 3.3 was modified to clarify the possible relation.

P12, line 31: "… ~ 40 observation hours.  Guyot et al. (2015) performed a similar experiment to investigate the sensitivity of the cloud spectrometers to meteorological parameters. Despite the fact that we conducted the measurements in different temperatures (in Puy-de-Dôme they sampled clouds only above zero) we found that our results were related. The main reason that caused discrepancies (mainly in deriving $N_c$ and *LWC*) to the fixed direction cloud spectrometers ground

setups (Pallas – CAPS and Puy-de-Dôme - FSSP) was the wind direction. The strong sensitivity to the wind direction suggested that the cloud spectrometers were sampling anisokinetically in both cases.

RC5:

*The authors go into great detail regarding alignment relative to the wind direction, and the discussion is rather laboured and lengthy. The effects on number concentration are not particularly surprising, but are elaborated in great detail, no doubt because the specific instrumental setups (e.g. the brake on the FSSP) require it. This discussion may benefit from being shortened.*

AA5:

As was discussed above, CAPS (our only fixed direction ground setup) was sensitive to the wind direction. Our main motivation during this section was to highlight this issue and explain our choice to limit our data and restrict them to isoaxial conditions when deriving $N_c$ and *LWC*. On the other hand, our detailed analysis indicated that derived parameters *ED* and *MVD* were not as sensitive to wind direction as $N_c$ and *LWC*. We would like to keep the detailed description and reasoning that support our results and conclusions even though they might seem to be lengthy for some readers. We also provide detailed guidelines on the data quality assessment since it is very hard to find it in literature.

However, we moved the detailed discussion on remained wind sectors of CAS and FSSP setups inter-comparison (p.12 line 4-25) to Supplementary Materials including Fig. 9 and 10.

RC6:

*I note the authors specifically mention the frequent occurrence of supercooled clouds at this location. Do they have further corroborative evidence that the clouds being sampled contained only supercooled liquid water drops. Whilst LWC is readily calculated in terms of the measured parameters, it would be useful if the authors could comment on whether any data relate ice particles.*

AA6:

We used three approaches to investigate ice particle content. Our analysis supports our claim on sampling mostly supercooled liquid water drops. Here, we will present a typical example of a cold day (15.11.2013 with temperature values around -10 °C).

First, the CAS Dpol depolarization features including particle-by-particle data were used to investigate asphericity of particles, similarly as described in detail by Meyer, (2012). For the detection of the particles asphericity, the polarized components of the scattered light are usually measured in backward direction because the scattering in that direction is influenced by the particle shape. In our case, during 15.11.2013, average value of the polarized component of the backscattered light for each particle was 0.27 (std 0.01). Meyer (2012) sampled natural clouds

while facing similar temperatures as we faced during PaCE. She explained in p. 74 "*the fraction of frozen cloud particles in the COALESC natural clouds is generally low. Especially above −13 ∘C/260 K, only few ice crystals are observed*".

Secondly, we performed Cloud Imaging Probe (CIP) data analysis and found that vast majority of the small drops in nonprecipitating clouds were spherical. However, we are familiar that spherical cloud droplets could also be connected with additional possible crystal rounding mechanisms (e.g. Nichmann et al, 2017)

Thirdly, we also used data from ceilometer that continuously measures at Kenttarova station - about 6 km downwind from Sammaltunturi station.

[Figure]

Sammaltunturi station altitude is 565m a.s.l. The highest values (purple) indicate liquid, low values (blue) are aerosol, and orange-red is snow. This day started off clear. Then, there was a supercooled liquid layer present close to the surface during the morning starting from just before 04:00. After 08:00, ice begins to precipitate through the layer, becoming stronger by midday. After about 17:00, this ice precipitation is becoming strong enough to almost fully glaciate the supercooled liquid layer, especially after 23:00. There, ice particles were expected to generate. However, the number of supercooled liquid droplets greatly exceed the number of small ice particles.

The possibility that we also sampled ice particles will be commented in results section of the revised manuscript.

P15, line 26: "…supercooled clouds. Although there is a possibility we sampled ice particles in some cases, it is expected that the number of supercooled liquid droplets greatly exceed the number of small ice cloud droplets"

RC7:

*Whilst the manuscript appears to be in scope for the journal, I would recommend revision before it could be considered for publication.*

AA7:

Major revision of the manuscript will be done according to both reviewers' recommendations.

RC8:

*On a technical level, I believe the quantity proi defined on p.7 line 22 should be the reciprocal of that displayed. Also, the quantities bi+1 should, I believe, be bi ∗+1.*

AA8:

The typo was corrected.

References

Guyot, G., Gourbeyre, C., Febvre, G., Shcherbakov, V., Burnet, F., Dupont, J.-C., Sellegri, K., and Jourdan, O.: Quantitative evaluation of seven optical sensors for cloud microphysical measurements at the Puy-de-Dôme Observatory, France, Atmos. 20 Meas. Tech., 8, 4347-4367, https://doi.org/10.5194/amt-8-4347-2015, 2015.

Meyer, J.: Ice Crystal Measurements with the New Particle Spectrometer NIXE-CAPS, Schriften des Forschungszentrum Jülich, Reihe Energie und Umwelt, 160, 2012 (https://core.ac.uk/download/pdf/34903548.pdf)

Nichman, L., Järvinen, E., Dorsey, J., Connolly, P., Duplissy, J., Fuchs, C., Ignatius, K., Sengupta, K., Stratmann, F., Möhler, O., Schnaiter, M., and Gallagher, M.: Intercomparison study and optical asphericity measurements of small ice particles in the CERN CLOUD experiment, Atmos. Meas. Tech., 10, 3231-3248, https://doi.org/10.5194/amt-10-3231-2017, 35 2017.

---

## Author Response (AR1)

The authors would like to thank both reviewers for their time, their helpful comments, suggestions and their attention to all the details. We appreciate their contribution. Please find below a detailed point-by-point replies and amendments followed by the marked up manuscript. Citations are listed at the end of the document. Changes to the manuscript are shown in red for deleted text and blue for added text.

**Reviewer 1**

Reviewer comment (RC)

Authors answer (AA)

RC1:

10 *The study that is discussed in this submission focuses on the performance of three cloud probes that were originally designed for operation on aircraft but an attempt has been made to adapt then to a ground based location where the environment can often be quite harsh with respect to icing conditions. I think that given the multiple issues with how the instruments were operated, this paper is overly lengthy. It could just as easily have been a very short technical note that points out how you should NOT mount and*
15 *operate instruments on the ground that are designed to be used for aircraft.*

AA1:

This aspect will be elaborated in the introduction and methods of the revised manuscript. There is lately an increased demand for long term continuous ground based in-situ cloud measurements. The approaches are solidified slowly but continuously. Unfortunately, there is more or less nonexistent instrumentation
20 to cover such demand, moreover continuous in-situ cloud measurements in conditions similar to those at our sub-Arctic location are very challenging. All three ground based setups were designed by the manufacturers (PSM and DMT). Our FSSP-100 ground set-up allows us to continuously follow the wind direction. CAPS was installed in fixed direction and we reported in our manuscript that wind direction was a crucial factor for its performance and caused artifacts in conditions with winds not iso-axial to
25 CAPS. However, the decision to use CAPS was due to two reasons. 1) To perform a full benchmarking and intercomparison with FSSP-100 and evaluate the results and 2) to quantify the usability of data that were not collected in the most favorable conditions – wind iso-axial direction. CAPS inhalation system (high flow pump and wide diameter hoses) is unfortunately so big that using rotational platform is not feasible. Our analysis clearly shows that for some parameters like number concentration and liquid water
30 content the iso-axial direction plays a significant role, but for the derived parameters effective diameter and median volume diameter it seems not to be disqualifying at all. This behavior of the instrument is extremely important to us because we are highly interested in semi and long-term continuous measurements. We do not consider this work just merely an instrument comparison but also as an experiment on how to operate the cloud probes to perform ground based measurements in harsh
35 environments.

The following text was amended

p.3 line 23:
"In this work, we focused on the intercomparison of three cloud spectrometer probes ground setups as they were used during the PaCE 2013. Due to the increased demand for long term continuous ground based in-situ cloud measurements, we tested and evaluated the operation of three FMI owned ground setups to perform continuous ground based measurements in harsh environments."

p.4, line 24:
"During PaCE 2013, to perform in-situ measurements of cloud droplets, we used three instruments originally developed for airborne measurements, but tailored for ground-based measurements by manufacturer (DMT, USA)."

p.12, line 11:
"In a similar way, all the remained sectors of the wind rose were investigated in detail to reveal more biases (for detailed description please see discussion and Fig. S5, S6 in the Supplement)"
The following figures and corresponding text was moved to SM

Fig.9, Fig.10 renamed Fig.S5, Fig.S6

p.12, line 13 -33.
"In a similar way, all the remained sectors of the wind rose were investigated in detail to reveal more biases. In Fig. 9 we summarized the most representative cases. Fig. 9a shows the whole wind iso-axial conditions sector as it was defined previously (200 – 235°) and ensures that there was good agreement ($R^2$ = 0.70 and slope 0.57). Fig. 9b shows that the CAS probe had more losses (factor from 3 to ~ 10) in $N_c$ when the wind direction was perpendicular to the CAS fixed direction, covering the sector from 115 to 154° ($R^2$ = 0.32 and slope 0.72). We also used observations when the wind direction ranged from 0 to 74° (Fig. 9c). There, due to the installation of the brake in FSSP' setup, an abnormality was created which clearly affected FSSP' ability to operate properly. The agreement between the two instruments in this sector of the wind rose was found the worst of all cases ($R^2$= 0.08 and slope 0.33). Finally (Fig. 9d), we used observations when the wind direction ranged from 95 to 114° in order to demonstrate one case when the wind direction was out of both, the wind iso-axial and perpendicular area. As expected, the CAS probe was affected by the wind direction. CAS was undercounting again when deriving $N_c$ (slightly less than in the case of perpendicular direction, $R^2$ = 0.54 and slope 0.64). Figure 10 presents the number size distributions for the same cases to investigate further the counting ability of the two instruments and find out the size bins where the probes had the biggest difference in counting. For size range from 1.2 to 7 μm, both cloud probes behaved the same in all wind directions. In Fig. 10a (200 – 235°) we noticed that the number size distribution in wind iso-axial case had only some minor differences in sizing (slight shift in FSSP sizing towards bigger sizes, about 1.5 μm) that were expected as we mentioned in the previous paragraph. In Fig. 10b (115 to 154°), where the wind was perpendicular to the CAS probe we lost a

significant number (maximum losses in counts up to 75%) of droplets in the size range from 8 to 30 µm. In Fig. 10c (0 to 74°), where the FSSP faced operational malfunction due to its brake installation setup, it undercounted cloud droplets (maximum losses in counts up to 85%) for sizes larger than 11.8 µm. Finally, in Fig 10d (95 to 114°) we observed that the behaviour of CAS was affected by the wind direction in a similar way as it was found for the perpendicular case. However, in this case CAS lost fewer droplets (maximum losses in counts up to 45% for size range from 8 to 30 µm)."

The following figures were deleted and replaced

Fig.14 and Fig.15 were deleted and replaced from Fig.12

RC2:

*What is puzzling is why the decision was made to use these instruments rather than the DMT Fog Monitor that utilizes the same measurement theory but was intentionally designed for ground based measurements*

AA2:

We are aware of the DMT Fog Monitor and its features and we agree with the reviewer that the Fog Monitor is the best choice for in situ ground based measurements, however only for warm clouds. Unfortunately, based on our experience, the fog monitor is not suitable for measurements in Pallas and similar environments. Spiegel et al. (2012) published a detailed investigation of the FM-100 (https://www.atmos-meas-tech.net/5/2237/2012/amt-5-2237-2012.pdf) where she analytically evaluated the instrument during its operation in Jungfraujoch and reports that FM-100 had issues in sub-zero temperatures due to icing conditions. There, the Fog Monitor showed several artifacts in temperatures below zero ( page 2239 Table 1 and page 2250 "*Due to the mounting position of the FM-100, the inlet often was completely closed by frozen cloud droplets as the cold and humid updraft blew into the inlet of the FM-100. We therefore excluded periods with temperatures below 0 ◦C from data evaluation in order to exclude potential measurement artifacts that might arise due to freezing*." Also, as we can see in fig 9d) in Spiegel et al. (2012), the median wind speed in Junfraujoch was less than 2 m/s, and 75th percentile less than 3 m/s. During PaCE, average wind speed was 6.8 m/s; increasing especially the aspiration-related losses at non-isoaxial conditions. High wind speeds with varying wind direction, together with sub-zero temperatures make the Fog Monitor unusable in conditions similar to those at our measurement site. We should also note that FM-100 was operated in Pallas during our cloud campaign in 2009, and the aforementioned issues prevented any decent utilization of the data (Spiegel, private communication 2020).

The following text was amended

p.3, line 9:

"Spiegel et al., (2012) made a thoroughly analysis of wind velocity and wind angle impacts at the Junfraujoch comparing the Fog droplet spectrometer (FM-100) to others instruments. FM-100 showed several artifacts at temperatures below zero"

RC3:

*They do not explain why they chose to mount and operate the instruments in the way they did instead of using a wind tunnel set up that would have circumvented the problems that arose.*
AA3:

Methods section will be revised to reflect reviewer comment. The reason behind our choice is that the atmospheric in-situ measurements community (in our case the European Research Infrastructure for the observation of Aerosol, Clouds and Trace Gases, ACTRIS) has identified these cloud droplet probes with surface installation as a potential method for continuous cloud in-situ measurements (ACTRIS-PPP Deliverable D5.1: Documentation on technical concepts and requirements for ACTRIS Observational Platforms,
https://www.actris.eu/Portals/46/Documentation/ACTRIS%20PPP/Deliverables/Public/WP5_D5.1_M18.pdf?ver=2018-06-28-125343-273 ). There are not ready standard operating procedures (SOP's) how to utilize these probes in continuous operation in practice, and our paper helps exactly towards this objective. Wind tunnel is a well-known approach and we agree with the reviewer that it might be considered as the optimal choice. Unfortunately, several measurement sites (e.g. in sub-arctic) do not have this possibility. This is due to both practical (e.g. our site is part of a natural park where big construction projects are prohibited) and budgetary reasons; Building a wind tunnel in many locations where cloud in-situ measurements could be conducted is not financially feasible.

The following text was added

p.4, line 23:
"The atmospheric in-situ measurements community has identified cloud droplet probes with surface installation as a potential method for continuous cloud in-situ measurements (Wandinger et al., 2018)."

p.4, line 29:
"Wind tunnel could be considered as the optimal choice to utilize these instruments for ground based setups (e.g Elk Mountain- Baumgardner et al., (1983) and Puy de Dôme- Guyot et al., (2015)). There are measurement sites like ours in sub-arctic which do not have this possibility due to both practical and budgetary reasons.  However, it was shown that same quality data could be obtained from roof top measurements (Guyot et al., 2015). Ground based measurements with cloud probes that were originally designed to be used for aircraft were already conducted in several measuring sites without using a wind tunnel (e.g. Jungfraujoch- Lloyed et al. (2015) and Storm Peak- Lowenthal et.al, (2019))."

RC4:

*As a three page note, there would be two principal conclusions: 1) Cloud probes with inlets should always be mounted into the wind and 2) proper deicing is always necessary when conditions dictate it.*

AA4:

Conclusions section will be modified accordingly to highlight the first conclusion. We agree that cloud probes with inlets should be mounted into the wind. Two of the ground setups (FSSP and CDP) that were installed and operated were following the wind direction and only CAPS was fixed to one direction. The role of the mutual direction of probe heading and the wind direction is one of our main conclusions (e.g. p16, line12, "*Results indicated that when we were deriving Nc, the mutual direction of probe heading and the wind direction were playing the most significant role. From the inter-comparison of the CAS (fixed orientation) against FSSP (rotating platform), it was found that the CAS probe had the best agreement ($R^2$ =0.70) with the FSSP during wind iso-axial conditions (200 to 235°). The CAS probe counting efficiency was strongly dependent on the wind direction, this can be clearly explained by its installation to fixed orientation*". However, we show that this was not valid when deriving effective diameter and median volume diameter.

FSSP and CAPS anti-ice feature was modified by manufacturer (DMT for CAPS and PMS for FSSP) for ground based measurements. CDP have used standard heating.

The following text was deleted

p.16, line 25 to p.17, line 2:

"Regarding the size distribution, we noticed some differences in our measurements. Even though all three probes were calibrated the same way, but each separately, we found that their sizing was slightly different in real atmospheric conditions. There was a slight shift in FSSP sizing towards bigger sizes in comparison to the CAS probe, ~1.5 µm in size range from 7 to 10 µm and a slight shift in CDP sizing towards smaller sizes in comparison with the CAS probe ~2.5 µm in size range from 5 to 7.5 µm. Our conclusions on the four derived parameters should take into account those sizing uncertainties. The FSSP, an instrument placed on rotational platform, with wider inlet opening of inhalation system, provided the best performance and data coverage for in-situ cloud droplets measurements. The CDP probe often accumulated ice in sub-zero condition, both in its rotational platform and inhalation system. This was happening due to presence of supercooled clouds at the station. The big surfaces of the CDPs rotation platform and the inlet with small opening were collecting ice very fast. However, when the station was in warm cloud and the temperature was above zero, CDP was operating well considering the cloud droplets counting."

p.16, line 31-33 were moved to p.17, line16:

" The CDP probe often accumulated ice in sub-zero condition, both in its rotational platform and inhalation system. This was happening due to presence of supercooled clouds at the station. The big surfaces of the CDPs rotation platform and the inlet with small opening were collecting ice very fast. "

The following text was added

p.17, line 32:

"At the time of PaCE2013, the market did not offer any instrumentation fulfilling our requirements on continuous long-term unattended operation at subzero conditions. As final suggestions regarding performing continuous ground based in-situ cloud measurements in harsh environments, we would like to highlight two major issues. First, the cloud probes should always continuously face the wind direction to minimize the sampling losses. If this is not secured, only the measurements that were conducted in wind iso-axial conditions can be used for further analysis. However, deriving the sizing parameter*s ED* and *MVD* for the whole wind direction spectrum is still possible, but must be done with insight and prudence. Secondly, deicing features of the ground setups should be upgraded to make possible their unattended operation in subzero conditions. Otherwise, the cloud probes need necessary daily or more frequent checkups and cleaning of their inlets. "

RC5:

*As a study comparing three instruments with similar measurement techniques it is not as useful or relevant as the number of other studies where instruments are compared in wind tunnels, such as the study done at Puy-de-Dôme referenced in the current study.*

AA5:

As we mentioned above in this text but also in the manuscript we do not compare the instruments themselves but three different setups developed for continuous in-situ cloud ground based measurements (e.g. p1, line15, "*The main motivation of the campaign was to conduct in-situ cloud measurements with three different cloud spectrometer probes and perform an evaluation of their ground based setups*"). We further clarified this aspect through the abstract of the revised manuscript.

The following text was amended

p1, line 18:
"We investigated how different meteorological parameters affect each instruments' ground based setup operation"

p1, line 20:

"we suggested limitations for further use of the instruments setups in campaigns where focus is on investigating aerosol cloud interactions"

p.1, line 24:

"A complete intercomparison between the CAS and the FSSP-100 ground setups and additionally between the FSSP-100 and the CDP ground setups was made and presented".

p.16, line 11:

"We deployed three cloud spectrometers' setups (CAS, FSSP and CDP) on the roof of Sammaltunturi station, located in Finnish sub-Arctic."

RC6:

*The current study would be much more useful and relevant if it encompassed not only the measurements at their site but discussed why they chose to operate their instruments as they did compared to other sites such as Storm Peak, Elk Mountain, Puyde-Dôme, Jungfraujoch, and the Zugspitze where similar studies have been done but more successfully. Storm Peak, Elk Mountain and Puy-de-Dôme all use a wind tunnel to introduce cloud air to the instruments so that the sensors are being under conditions more like they were designed for, i.e. aircraft.*

AA6:

We thank the reviewer for this comment and as mentioned above we agree that wind tunnel might be considered as an optimal choice and colleagues at those sites did a great job. Methods section will be revised to explain our choice. References from measuring sites where they use ground based setup of the cloud probes without using a wind tunnel will be added in the revised manuscript. We deeply believe that ground based setup with inhalation system can also be considered as an acceptable approach. Further, there exist several studies where researchers performed ground based measurements with cloud probes without using a wind tunnel. However, we would like to highlight that our main goal is not to show that our approach is the best for ground based measurements (we don't aim to compare ways of using the instruments in several measurement sites) but that it is an approach which is suitable in conditions where options are limited.

Puy de-Dome is a site where we several times contributed during intercomparison campaigns and we are aware of the way they perform cloud measurements. They obtained data not only from the wind tunnel but also from the roof top. FSSP-100 they use on the roof top was fixed or manually rotated (our FSSP-100 setup follows continuously the wind direction without need of manpower). In the study done at Puy-de-Dôme we referenced (Guyot et al.2015, https://doi.org/10.5194/amt-8-4347-2015) we can see in Fig.2 in there that when the wind direction was favorable **both approaches (wind tunnel and roof top mount) provide data of the same quality**. Also we can see in Lowenthal et. al 2019 (https://www.atmos-chemphys.net/19/5387/2019/acp-19-5387-2019.pdf), that in Storm Peak, ground based cloud measurements were conducted without using a wind tunnel (Fig. 1 in there). They explain in p.5389 that "*The cloud probes were mounted on a rotating wind vane (to orient them into the wind) located on the west (upwind) railing of the roof approximately 6 m above the snow surface*"). In addition, they highlight the need for higher resolution instruments for distinguish between liquid and ice particles in mixed phase clouds in p.5399 "*They also demonstrate the limitations of instrumentation such as the FSSP-100 and CIP (2-D optical array probe) for distinguishing liquid droplets from small ice crystals in mixed phase clouds. Higher-resolution instruments are required for this purpose.*" Their statement highlights the importance of CAPS ground setup due to CAS depolarization features (note: not a subject of our current manuscript). Finally, Lloyed et al., (2015), during the Cloud Aerosol Characterisation Experiments (CLACE) and the Ice Nucleation Process Investigation and Quantification project (INUPIAQ) in Jungfraujoch also used cloud probes for ground based measurements without using a wind tunnel. (Fig. 2 in there) p12954 "*An overview of relevant instrumentation at this site can be found in Table 1, and some of these instruments (that were mounted on a pan and tilt rotator wing) are labelled in Fig. 2. The rotator allowed us to automatically adjust the position of the instruments based on information about the wind direction and vertical wind angle from a sonic anemometer*".

Text that was added corresponding to RC6 could be found below together with text that was added corresponding to RC7.

RC7:
*I can't recommend this manuscript for publication in its present form as I don't find the results that useful other than as a warning about how not to operate these instruments. A more comprehensive review of ground based measurements with sensors designed for aircraft would be far more useful.*
*Although I am the chief scientist and founder of Droplet Measurement Technologies, I was not involved with the setting up of the instruments that were involved in this study or the ventilation systems used to introduce cloud air. I have tried to ascertain how this all evolved but the technical staff who were involved are no longer with the company so I have no way of understanding the history of this project. I would recommend that the authors consider a different approach for future studies.*

AA7:

As it was discussed already above, our aim is long term continuous in-situ ground based measurements and unfortunately current cloud probes do not fulfill such requirements. They are difficult to be installed facing automatically towards the wind due to their shape. Results section will be modified to note the need for new instrumentation to fulfill those requirements.

All three cloud probes (FSSP, CDP and CAPS) and their setups discussed in this manuscript were designed and sold to FMI as "in-situ cloud ground based measurement setups" by the manufacturers (PSM and DMT).

The following text was added (corresponding to RC6 and RC7)

p.3, line 14:
"They placed one FSSP and the fog monitor at the roof of the observatory and the two CDP probes and one FSSP inside a wind tunnel."

p.3, line 16:
"Lloyed et al. (2015) observed cloud microphysical structures by conducting CAPS, FSSP, CDP-100 and PVM measurements. They mounted all the instruments on a rotator and wing on the terrace rooftop outside the Sphinx Laboratory, (Jungfraujoch, Switzerland)."

p.3, line 20:
"Lowenthal et.al, (2019) conducted winter time mixed- phase orographic cloud measurements at the Storm Peak Laboratory (Colorado, USA). They deployed a FSSP-100 forward-scattering spectrometer probe on a rotating wind vane (to orient them into the wind)."

p.4, line 29:
"Wind tunnel could be considered as the optimal choice to utilize these instruments for ground based setups (e.g Elk Mountain= Baumgardner et al., (1983) and Puy de Dôme - Guyot et al., 2015). There are measurement sites like ours in sub-arctic which do not have this possibility due to both practical and budgetary reasons. However, it was shown that same quality data could be obtained from roof top measurements (Guyot et al., 2015). Performing measurements with cloud probes that were originally manufactured for airborne measurement without using a wind tunnel were already conducted in several measuring sites (e.g. Jungfraujoch- Lloyed et al. (2015) and Storm Peak- Lowenthal et.al, (2019))."

p.13, line 15:
"Guyot et al. (2015) performed a similar experiment to investigate the sensitivity of the FSSP to meteorological parameters. Even though we conducted the measurements at different temperatures (in Puy-de-Dôme they sampled clouds only above zero) we found that our results were related. The main reason that caused the discrepancies (mainly in deriving $N_c$ and $LWC$) to the fixed direction cloud spectrometers ground setups (Pallas – CAPS and Puy-de-Dôme - FSSP) was the wind direction. The strong sensitivity to the wind direction suggested that the cloud spectrometers were sampling anisokinetically in both cases"

**Reviewer 2**

RC1:

*Whilst the title suggests it is an intercomparison between these instruments, it is really a comparison of the rather specific experimental setups employed during this campaign. The paper therefore serves more as a campaign report than a reference on how best to set up experiments for ground-based in-situ cloud measurement. Nevertheless, it does highlight some of the pitfalls.*

AA1:

We agree with the reviewer that this was an intercomparison between the specific ground setups and not the instruments themselves and we clarified this through the manuscript (e.g. p1, line 15, "The main motivation of the campaign was to conduct in-situ cloud measurements with three different cloud spectrometer probes and perform an evaluation of their ground based setups"). We consider that this work was not just an intercomparison but also an operative experiment on how to operate cloud probes for ground based measurements during harsh conditions. We will change the manuscript title to avoid any misunderstandings and possible confusions to: "In-situ cloud ground based measurements in Finnish sub-Arctic: Intercomparison of three cloud spectrometers setups ". Also, we clarified in the abstract of the revised manuscript that we intercompared the experimental ground setups of the cloud probes.

The manuscript title was amended

"In-situ cloud ground based measurements in Finnish sub-Arctic: Intercomparison of three cloud spectrometers setups "

The following text was amended

p.1, line 18:
"We investigated how different meteorological parameters affect each instruments' ground based setup operation"

p.1, line 20:
"we suggested limitations for further use of the instruments setups in campaigns where focus is on investigating aerosol cloud interactions"

p.1, line 24:
"A complete intercomparison between the CAS and the FSSP-100 ground setups and additionally between the FSSP-100 and the CDP ground setups was made and presented".

p.3, line 23:

"In this work, we focused on the intercomparison of three cloud spectrometer probes ground setups as they were used during the PaCE 2013"

p.16, line 11:

"We deployed three cloud spectrometers' setups (CAS, FSSP and CDP) on the roof of Sammaltunturi station, located in Finnish sub-Arctic."

The following text was added

p.3, line 24:

"Due to the increased demand for long term continuous ground based in-situ cloud measurements, we tested and evaluated the operation of three FMI owned ground setups to perform continuous ground based measurements in harsh environments."

RC2:

*I am aware of the motivation within communities such as ACTRIS for the establishment of long term ground-based in-situ cloud measurements, and as such this paper is a step in the right direction with respect to evaluating how these might be established. However, the conclusions do not seem robust enough to form the basis of wider recommendations. The paper does highlight the considerable difficulties faced by any attempt at long-term observations, and it is evident that any plans for unattended operation would pose particular challenges, especially in the sub-Arctic environment. In fairness, the authors are conservative in their recommendations and focus on Pallas campaigns (past analysis and future experiments).*

AA2:

We thank the reviewer acknowledging the demand within community for long-term ground-based in-situ cloud measurements and understanding the main motivation of our work. We indeed are conservative and focus on Pallas campaigns. Our recommendations are based on results we obtained from continuous (about two and a half months) PaCE campaign at harsh sub-Arctic conditions. There are two main conclusions that we highlight as basis for wider recommendations. Conclusions were modified accordingly:

The following text was added

p.17, line 32:

"At the time of PaCE2013, the market did not offer any instrumentation fulfilling our requirements on continuous long-term unattended operation at subzero conditions. As final suggestions regarding performing continuous ground based in-situ cloud measurements in harsh environments, we would like to highlight two major issues. First, the cloud probes should always continuously face the wind direction to minimize the sampling losses. If this is not secured, only the measurements that were conducted in

wind iso-axial conditions can be used for further analysis. However, deriving the sizing parameter*s ED* and *MVD* for the whole wind direction spectrum is still possible, but must be done with insight and prudence. Secondly, deicing features of the ground setups should be upgraded to make possible their unattended operation at subzero conditions. Otherwise, the cloud probes need necessary daily or more frequent checkups and cleaning of their inlets.

The following text was deleted

p.16, line 25  to p.17, line 2

"Regarding the size distribution, we noticed some differences in our measurements. Even though all three probes were calibrated the same way, but each separately, we found that their sizing was slightly different in real atmospheric conditions. There was a slight shift in FSSP sizing towards bigger sizes in comparison to the CAS probe, ~1.5 µm in size range from 7 to 10 µm and a slight shift in CDP sizing towards smaller sizes in comparison with the CAS probe ~2.5 µm in size range from 5 to 7.5 µm. Our conclusions on the four derived parameters should take into account those sizing uncertainties. The FSSP, an instrument placed on rotational platform, with wider inlet opening of inhalation system, provided the best performance and data coverage for in-situ cloud droplets measurements. The CDP probe often accumulated ice in sub-zero condition, both in its rotational platform and inhalation system. This was happening due to presence of supercooled clouds at the station. The big surfaces of the CDPs rotation platform and the inlet with small opening were collecting ice very fast. However, when the station was in warm cloud and the temperature was above zero, CDP was operating well considering the cloud droplets counting."

p.16, line 31-33 were moved to p.17, line16.

" The CDP probe often accumulated ice in sub-zero condition, both in its rotational platform and inhalation system. This was happening due to presence of supercooled clouds at the station. The big surfaces of the CDPs rotation platform and the inlet with small opening were collecting ice very fast. "

RC3:

*I note that this experiment was performed contemporaneously with that at Puy-de Dôme (Guyot et al, 2015) and hence the insight from the findings of the latter were not available to provide guidance on what complementary instruments, based on ensembles of particles, might be installed and used to explore scaling of the number concentration and related parameters. In fairness, the authors do recommend such instrumentation for future campaigns.*

AA3: We agree with the reviewer and we indeed recommend that such instrumentation can be really useful for future or similar campaigns, especially for measuring LWC (p13, line 22 ". In addition, we suggest the deployment of another LWC sensor, e.g. the particle volume monitor (PVM-100, Gerber 1999) during future campaigns in order to obtain another reference LWC values for inter-comparison in wide temperature range").

The following text was added

p.14, line 11:
"In addition, we are continuously following the development of a new generation of counters designed for ground based in-situ cloud measurements. Thus, it is a matter of future deployment during upcoming PaCE campaigns. "

RC4:

*Guyot et al (2015) found that FSSP measurements suggested anisokinetic sampling and a high sensitivity to the wind speed and direction. It would be helpful if the authors could comment on how their findings relate to this earlier analysis.*

AA4:

 According to Guyot et.al. (2015) for wind speeds larger than 3 m/s, the sensitivity of FSSP to the wind direction was high (wind speed average value in Pallas was 6.8 m/s) (page4360 in Guyot et al (2015) "On average, the greater the angular deviation from isoaxial configuration is, the more the size distribution is reduced, except for a 3 m/s wind speed"). We agree that this was the main reason that caused discrepancies to the fixed direction of CAPS cloud probe ground setup (the only instrument with fixed direction, discussed in detail (section 3.3)) and it mainly affected its size distribution and hence the number concentration. In our case FSSP was continuously following the wind direction without the need of man force in comparison with Puy-de-Dôme. For this reason, FSSP sensitivity was mainly connected with its brake installation and not the anisokinetic sampling. We should also highlight that in comparison to Guyot et.al. (2015) where they conducted measurements in temperatures above zero, we were usually facing temperature below zero. In revised manuscript section 3.3 was modified to clarify the possible relation.

The following text was added

p.13, line 7:
"Guyot et al. (2015) performed a similar experiment to investigate the sensitivity of the FSSP to meteorological parameters. Even though we conducted the measurements at different temperatures (in Puy-de-Dôme they sampled clouds only above zero) we found that our results were related. The main reason that caused the discrepancies (mainly in deriving $N_c$ and $LWC$) to the fixed direction cloud spectrometers ground setups (Pallas – CAPS and Puy-de-Dôme - FSSP) was the wind direction. The strong sensitivity to the wind direction suggested that the cloud spectrometers were sampling anisokinetically in both cases."
RC5:

*The authors go into great detail regarding alignment relative to the wind direction, and the discussion is rather laboured and lengthy. The effects on number concentration are not particularly surprising, but are*

*elaborated in great detail, no doubt because the specific instrumental setups (e.g. the brake on the FSSP) require it. This discussion may benefit from being shortened.*

AA5:

As was discussed above, CAPS (our only fixed direction ground setup) was sensitive to the wind direction. Our main motivation during this section was to highlight this issue and explain our choice to limit our data and restrict them to isoaxial conditions when deriving Nc and LWC. On the other hand, our detailed analysis indicated that derived parameters ED and MVD were not as sensitive to wind direction as Nc and LWC. We would like to keep the detailed description and reasoning that support our results and conclusions even though they might seem to be lengthy for some readers. We also provide detailed guidelines on the data quality assessment since it is very hard to find it in literature.

However, we moved the detailed discussion on remained wind sectors of CAS and FSSP setups inter-comparison (p.12 line 4-25) to Supplementary Materials (SM) including Fig. 9 and 10.

The following figures and corresponding text was moved to SM

Fig.9, Fig.10 renamed Fig.S5, Fig.S6

p.12, line 13 -33.
"In a similar way, all the remained sectors of the wind rose were investigated in detail to reveal more biases. In Fig. 9 we summarized the most representative cases. Fig. 9a shows the whole wind iso-axial conditions sector as it was defined previously (200 – 235°) and ensures that there was good agreement ($R^2 = 0.70$ and slope 0.57). Fig. 9b shows that the CAS probe had more losses (factor from 3 to ~ 10) in $N_c$ when the wind direction was perpendicular to the CAS fixed direction, covering the sector from 115 to 154° ($R^2 = 0.32$ and slope 0.72). We also used observations when the wind direction ranged from 0 to 74° (Fig. 9c). There, due to the installation of the brake in FSSP' setup, an abnormality was created which clearly affected FSSP' ability to operate properly. The agreement between the two instruments in this sector of the wind rose was found the worst of all cases ($R^2 = 0.08$ and slope 0.33). Finally (Fig. 9d), we used observations when the wind direction ranged from 95 to 114° in order to demonstrate one case when the wind direction was out of both, the wind iso-axial and perpendicular area. As expected, the CAS probe was affected by the wind direction. CAS was undercounting again when deriving $N_c$ (slightly less than in the case of perpendicular direction, $R^2 = 0.54$ and slope 0.64). Figure 10 presents the number size distributions for the same cases to investigate further the counting ability of the two instruments and find out the size bins where the probes had the biggest difference in counting. For size range from 1.2 to 7 µm, both cloud probes behaved the same in all wind directions. In Fig. 10a (200 – 235°) we noticed that the number size distribution in wind iso-axial case had only some minor differences in sizing (slight shift in FSSP sizing towards bigger sizes, about 1.5 µm) that were expected as we mentioned in the previous paragraph. In Fig. 10b (115 to 154°), where the wind was perpendicular to the CAS probe we lost a

significant number (maximum losses in counts up to 75%) of droplets in the size range from 8 to 30 µm. In Fig. 10c (0 to 74°), where the FSSP faced operational malfunction due to its brake installation setup, it undercounted cloud droplets (maximum losses in counts up to 85%) for sizes larger than 11.8 µm. Finally, in Fig 10d (95 to 114°) we observed that the behaviour of CAS was affected by the wind direction in a similar way as it was found for the perpendicular case. However, in this case CAS lost fewer droplets (maximum losses in counts up to 45% for size range from 8 to 30 µm)."

RC6:

*I note the authors specifically mention the frequent occurrence of supercooled clouds at this location. Do they have further corroborative evidence that the clouds being sampled contained only supercooled liquid water drops. Whilst LWC is readily calculated in terms of the measured parameters, it would be useful if the authors could comment on whether any data relate ice particles.*

AA6:

We used three approaches to investigate ice particle content. Our analysis supports our claim on sampling mostly supercooled liquid water drops. Here, we will present a typical example of a cold day (15.11.2013 with temperature values around -10 ºC).

First, the CAS Dpol depolarization features including particle-by-particle data were used to investigate asphericity of particles, similarly as described in detail by Meyer, (2012). For the detection of the particles asphericity, the polarized components of the scattered light are usually measured in backward direction because the scattering in that direction is influenced by the particle shape. In our case, during 15.11.2013, average value of the polarized component of the backscattered light for each particle was 0.27 (std 0.01). Meyer (2012) sampled natural clouds while facing similar temperatures as we faced during PaCE. She explained in p. 74 "the fraction of frozen cloud particles in the COALESC natural clouds is generally low. Especially above −13 ℃/260 K, only few ice crystals are observed".

Secondly, we performed Cloud Imaging Probe (CIP) data analysis and found that vast majority of the small drops in nonprecipitating clouds were spherical. However, we are familiar that spherical cloud droplets could also be connected with additional possible crystal rounding mechanisms (e.g. Nichmann et al, 2017)

Thirdly, we also used data from ceilometer that continuously measures at Kenttarova station - about 6 km downwind from Sammaltunturi station.

[Figure]

Sammaltunturi station altitude is 565m a.s.l. The highest values (purple) indicate liquid, low values (blue) are aerosol, and orange-red is snow. This day started off clear. Then, there was a supercooled liquid layer present close to the surface during the morning starting from just before 04:00. After 08:00, ice begins to precipitate through the layer, becoming stronger by midday. After about 17:00, this ice precipitation is becoming strong enough to almost fully glaciate the supercooled liquid layer, especially after 23:00. There, ice particles were expected to generate. However, the number of supercooled liquid droplets greatly exceed the number of small ice particles.

The possibility that we also sampled ice particles was commented in results section of the revised manuscript. Such analysis is however out of the scope of current manuscript and thus will not be discussed within the manuscript.

The following text was added

p.16, line 16:
"Although there is a possibility we sampled ice particles in some cases, it is expected that the number of supercooled liquid droplets greatly exceed the number of small ice cloud droplets"
RC7:

*Whilst the manuscript appears to be in scope for the journal, I would recommend revision before it could be considered for publication.*

AA7:

Major revision of the manuscript will be done according to both reviewers' recommendations.

RC8:

*On a technical level, I believe the quantity proi defined on p.7 line 22 should be the reciprocal of that displayed. Also, the quantities bi+1 should, I believe, be bi ∗ +1.*

AA8:

The typo was corrected.

The following text was amended

p.8 line 1:

"where $pro_{i^*} = \frac{LWC}{LWC_i}$ is the proportion of the spectrum $LWC$ that falls in the $i$-*th* bin and

$cum_{i^*} = pro_1 + \cdots + pro_{i^*}$ is the cumulative proportion of the spectrum $LWC$ that falls in the first $i$ bins and

$i^*$ is the smallest value of $i$ such that $cum_{i^*} > 0.5$."

References (with blue color those that were added in revised manuscript)

Wandinger U., Apituley A., Blumenstock T., Bukowiecki N., Cammas J.-P., Connolly P., De Mazière M., Dils B., Fiebig M., Freney E., Gallagher M., Godin-Beekmann S., Goloub P., Gysel M., Haeffelin M., Hase F., Hermann M., Herrmann H., Jokinen T., Komppula M., Kubistin D., Langerock B., Lihavainen H., Mihalopoulos N., Laj P., Lund Myhre C., Mahieu E., Mertes ., Möhler O., Mona L., Nicolae D., O'Connor E., Palm M., Pappalardo G., Pazmino A., Petäjä T., Philippin S., Plass-Duelmer C., Pospichal B., Putaud J.-P., Reimann S., Rohrer F., Russchenberg H., Sauvage S., Sellegri K., Steinbrecher R., Stratmann F., Sussmann R., Van Pinxteren D., Van Roozendael M., Vigouroux C., Walden C., Wegene R. and Wiedensohler A. ACTRIS-PPP Deliverable D5.1: Documentation on technical concepts and requirements for ACTRIS Observational Platforms,https://www.actris.eu/Portals/46/Documentation/ACTRIS%20PPP/Deliverables/Public/WP5_D5.1_M18.pdf?ver=2018-06-28-125343-273 , 2018.

Spiegel, J. K., Zieger, P., Bukowiecki, N., Hammer, E., Weingartner, E., and Eugster, W.: Evaluating the capabilities and uncertainties of droplet measurements for the fog droplet spectrometer (FM-100), Atmos. Meas. Tech., 5, 2237–2260, doi:10.5194/amt5-2237-2012, 2012.

Guyot, G., Gourbeyre, C., Febvre, G., Shcherbakov, V., Burnet, F., Dupont, J.-C., Sellegri, K., and Jourdan, O.: Quantitative evaluation of seven optical sensors for cloud microphysical measurements at

the Puy-de-Dôme Observatory, France, Atmos. Meas. Tech., 8, 4347-4367, https://doi.org/10.5194/amt-8-4347-2015, 2015.

Lowenthal, D. H., Hallar, A. G., David, R. O., McCubbin, I. B., Borys, R. D., and Mace, G. G.: Mixed-phase orographic cloud microphysics during StormVEx and IFRACS, Atmos. Chem. Phys., 19, 5387–5401, https://doi.org/10.5194/acp-19-5387-2019, 2019.

Lloyd, G., Choularton, T. W., Bower, K. N., Gallagher, M. W., Connolly, P. J., Flynn, M., Farrington, R., Crosier, J., Schlenczek, O., Fugal, J., and Henneberger, J.: The origins of ice crystals measured in mixed-phase clouds at the high-alpine site Jungfraujoch, Atmos. Chem. Phys., 15, 12953–12969, https://doi.org/10.5194/acp-15-12953-2015, 2015.

Meyer, J.: Ice Crystal Measurements with the New Particle Spectrometer NIXE-CAPS, Schriften des Forschungszentrum Jülich, Reihe Energie und Umwelt, 160, 2012 (https://core.ac.uk/download/pdf/34903548.pdf)

[revised manuscript text omitted]